# Genetic history from the Middle Neolithic to present on the Mediterranean island of Sardinia

Joseph H. Marcus [ID] et al.#

The island of Sardinia has been of particular interest to geneticists for decades. The current model for Sardinia's genetic history describes the island as harboring a founder population that was established largely from the Neolithic peoples of southern Europe and remained isolated from later Bronze Age expansions on the mainland. To evaluate this model, we generate genome-wide ancient DNA data for 70 individuals from 21 Sardinian archaeological sites spanning the Middle Neolithic through the Medieval period. The earliest individuals show a strong affinity to western Mediterranean Neolithic populations, followed by an extended period of genetic continuity on the island through the Nuragic period (second millennium BCE). Beginning with individuals from Phoenician/Punic sites (first millennium BCE), we observe spatially-varying signals of admixture with sources principally from the eastern and northern Mediterranean. Overall, our analysis sheds light on the genetic history of Sardinia, revealing how relationships to mainland populations shifted over time.

The whole-genome sequencing in 2012 of "Ötzi", an individual who was preserved in ice for over 5000 years near the Italo-Austrian border, revealed a surprisingly high level of shared ancestry with present-day Sardinian individuals[1,2]. Subsequent work on genome-wide variation in ancient Europeans found that most "early European farmer" individuals, even when from geographically distant locales (e.g., from Sweden, Hungary and Spain) have their highest genetic affinity with present-day Sardinian individuals[3–6]. Accumulating ancient DNA (aDNA) results have provided a framework for understanding how early European farmers show such genetic affinity to modern Sardinians.

In this framework, Europe was first inhabited by Paleolithic and later Mesolithic hunter-gatherer groups. Then, starting about 7000 BCE, farming peoples arrived from the Middle East as part of a Neolithic transition[7], spreading through Anatolia and the Balkans[8,9] while progressively admixing with local hunter-gatherers[10]. Major movements from the Eurasian Steppe, beginning about 3000 BCE, resulted in further admixture throughout Europe[11–14]. These events are typically modeled in terms of three ancestry components: western hunter gatherers ("WHG"), early European farmers ("EEF"), and Steppe pastoralists ("Steppe"). Within this broad framework, the island of Sardinia is thought to have received a high level of EEF ancestry early on and then remained mostly isolated from the subsequent admixture occurring on mainland Europe[1,2]. However, this specific model for Sardinian population history has not been tested with genome-wide aDNA data from the island.

The oldest known human remains on Sardinia date to ~20,000 years ago[15]. Archeological evidence suggests Sardinia was not densely populated in the Mesolithic, and experienced a population expansion coinciding with the Neolithic transition in the sixth millennium BCE[16]. Around this time, early Neolithic pottery assemblages were spreading throughout the western Mediterranean, including Sardinia, in particular vessels decorated with Cardium shell impressions (variably described as Impressed Ware, Cardial Ware, Cardial Impressed Ware)[17], with radiocarbon dates indicating a rapid westward maritime expansion around 5500 BCE[18]. In the later Neolithic, obsidian originating from Sardinia is found throughout many western Mediterranean archeological sites[19], indicating that the island was integrated into a maritime trade network. In the middle Bronze Age, about 1600 BCE, the "Nuragic" culture emerged, named for the thousands of distinctive stone towers, known as *nuraghi*[20]. During the late Nuragic period, the archeological and historical record shows the direct influence of several major Mediterranean groups, in particular the presence of Mycenaean, Levantine and Cypriot traders. The Nuragic settlements declined throughout much of the island as, in the late 9th and early 8th century BCE, Phoenicians originating from present-day Lebanon and northern Palestine established settlements concentrated along the southern shores of Sardinia[21]. In the second half of the 6th century BCE, the island was occupied by Carthaginians (also known as Punics), expanding from the city of Carthage on the North-African coast of present-day Tunisia, which was founded in the late 9th century by Phoenicians[22,23]. Sardinia was occupied by Roman forces in 237 BCE, and turned into a Roman province a decade later[24]. Throughout the Roman Imperial period, the island remained closely aligned with both Italy and central North Africa. After the fall of the Roman empire, Sardinia became increasingly autonomous[24], but interaction with the Byzantine Empire, the maritime republics of Genova and Pisa, the Catalan and Aragonese Kingdom, and the Duchy of Savoy and Piemonte continued to influence the island[25,26].

The population genetics of Sardinia has long been studied, in part because of its importance for medical genetics[27,28]. Pioneering studies found evidence that Sardinia is a genetic isolate with appreciable population substructure[29–31]. Recently, Chiang et al.[32] analyzed whole-genome sequences[33] together with continental European aDNA. Consistent with previous studies, they found the mountainous Ogliastra region of central/eastern Sardinia carries a signature of relative isolation and subtly elevated levels of WHG and EEF ancestry.

Four previous studies have analyzed aDNA from Sardinia using mitochondrial DNA. Ghirotto et al.[34] found evidence for more genetic turnover in Gallura (a region in northern Sardinia with cultural/linguistic connections to Corsica) than Ogliastra. Modi et al.[35] sequenced mitogenomes of two Mesolithic individuals and found support for a model of population replacement in the Neolithic. Olivieri et al.[36] analyzed 21 ancient mitogenomes from Sardinia and estimated the coalescent times of Sardinian-specific mtDNA haplogroups, finding support for most of them originating in the Neolithic or later, but with a few coalescing earlier. Finally, Matisoo-Smith et al.[37] analyzed mitogenomes in a Phoenician settlement on Sardinia and inferred continuity and exchange between the Phoenician population and broader Sardinia. One additional study recovered β-thalessemia variants in three aDNA samples and found one carrier of the cod39 mutation in a necropolis used in the Punic and Roman periods[38]. Despite the initial insights these studies reveal, none of them analyze genome-wide autosomal data, which has proven to be useful for inferring population history[39].

Here, we generate genome-wide data from the skeletal remains of 70 Sardinian individuals radiocarbon dated to between 4100 BCE and 1500 CE. We investigate three aspects of Sardinian population history: First, the ancestry of individuals from the Sardinian Neolithic (ca. 5700–3400 BCE)—who were the early peoples expanding onto the island at this time? Second, the genetic structure through the Sardinian Chalcolithic (i.e., Copper Age, ca. 3400–2300 BCE) to the Sardinian Bronze Age (ca. 2300–1000 BCE)—were there genetic turnover events through the different cultural transitions observed in the archeological record? And third, the post-Bronze Age contacts with major Mediterranean civilizations and more recent Italian populations—have they resulted in detectable gene flow?

Our results reveal insights about each of these three periods of Sardinian history. Specifically, our earliest samples show affinity to the early European farmer populations of the mainland, then we observe a period of relative isolation with no significant evidence of admixture through the Nuragic period, after which we observe evidence for admixture with sources from the northern and eastern Mediterranean.

## Results

**Ancient DNA from Sardinia**. We organized a collection of skeletal remains (Supp. Fig. 1) from (1) a broad set of previously excavated samples initially used for isotopic analysis[40], (2) the Late Neolithic to Bronze Age Seulo cave sites of central Sardinia[41], (3) the Neolithic Sites Noedalle and S'isterridolzu[42], (4) the Phoenician-Punic sites of Monte Sirai[23] and Villamar[43], (5) the Imperial Roman period site at Monte Carru (Alghero)[44], (6) medieval remains from the site of Corona Moltana[45], (7) medieval remains from the necropolis of the Duomo of San Nicola[46]. We sequenced DNA libraries enriched for the complete mitochondrial genome as well as a targeted set of 1.2 million single nucleotide polymorphisms (SNPs)[47]. After quality control, we arrived at a final set of 70 individuals with an average coverage of 1.02× at targeted SNPs (ranging from 0.04× to 5.39× per individual) and a median number of 466,049 targeted SNPs covered at least once per individual. We obtained age estimates by either direct radiocarbon dating (n = 53), previously reported

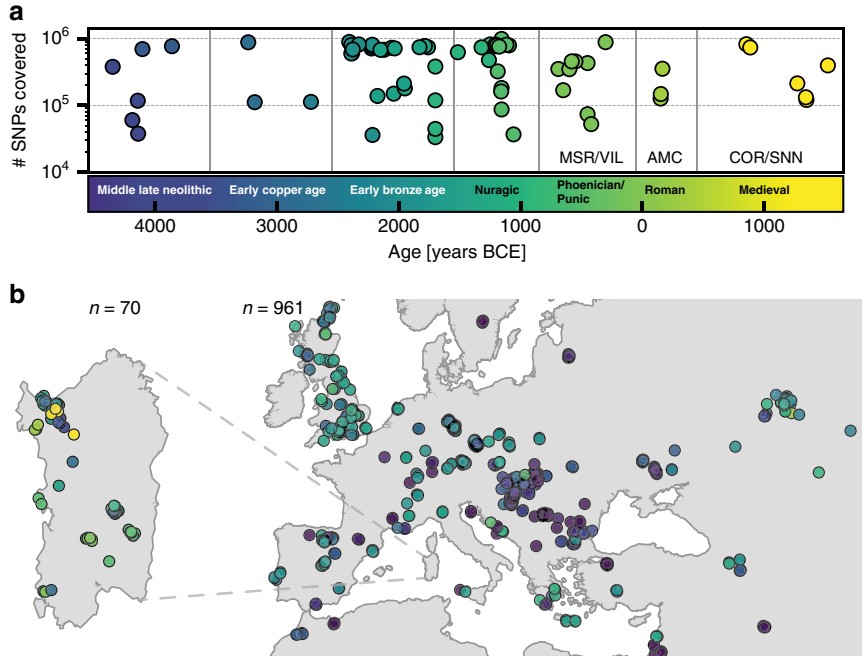

**Fig. 1 Number of SNPs covered, sampling locations and ages of ancient individuals. a** The number of SNPs covered at least once and age (mean of $2\sigma$ radiocarbon age estimates) for the 70 ancient Sardinian individuals. **b** The sampling locations of ancient Sardinian individuals and a reference dataset of 961 ancient individuals from across western Eurasia and North Africa (with "jitter" added to prevent overplotting; see Supp Data 1E for exact locations).

radiocarbon dates ($n = 13$), or archeological context and radio-carbon dates from the same burial site ($n = 4$). The estimated ages range from 4100 years BCE to 1500 years CE (Fig. 1, Supp. Data 1A). We pragmatically grouped the data into broad periods: Middle/Late Neolithic ('Sar-MN', 4100–3500 BCE, $n = 6$), Early Copper Age ('Sar-ECA', 3500–2500 BCE, $n = 3$), Early Middle Bronze Age ('Sar-EMBA', 2500–1500 BCE, $n = 27$), and Nuragic ('Sar-Nur', 1500–900 BCE, $n = 16$). For the post-Nuragic sites, there is substantial genetic heterogeneity within and among sites, and so we perform analysis per site when grouping is necessary ('Sar-MSR' and 'Sar-VIL' for the Phoenician and Punic sites of Monte Sirai, $n = 2$; and Villamar, $n = 6$; 'Sar-ORC002' for a Punic period individual from the interior site of S'Orcu 'e Tueri, $n = 1$; 'Sar-AMC' for the Roman period site of Monte Carru near Alghero, $n = 3$; 'Sar-COR' for the early medieval individuals from the site of Corona Moltana, $n = 2$; and 'Sar-SNN' for the medieval San Nicola necropoli, $n = 4$). Figure 1 provides an overview of the sample.

To assess the relationship of the ancient Sardinian individuals to other ancient and present-day west Eurasian and North-African populations we analyzed our individuals alongside published autosomal DNA data (ancient: 972 individuals[9,10,13,48–50]; modern: 1963 individuals from outside Sardinia[7] and 1577 individuals from Sardinia[32,33]). For some analyses, we grouped the modern Sardinian individuals into eight geographic regions (see inset in panel c of Fig. 2 for listing and abbreviations, also see Supp. Data 1E) and for others we subset the more isolated Sardinian region of Ogliastra ('Sar-Ogl', $n = 419$) and the remainder ('Sar-non Ogl', $n = 1158$). As with other human genetic variation studies, population annotations are important to consider in the interpretation of results.

**Similarity to western mainland Neolithic populations**. We found low differentiation between Middle/Late Neolithic Sardinian individuals and Neolithic western mainland European populations, in particular groups from Spain (Iberia-EN) and

southern France (France-N). When projecting ancient individuals onto the top two principal components (PCs) defined by modern variation, the Neolithic ancient Sardinian individuals sit between early Neolithic Iberian and later Copper Age Iberian populations, roughly on an axis that differentiates WHG and EEF populations, and embedded in a cluster that additionally includes Neolithic British individuals (Fig. 2). This result is also evident in terms of genetic differentiation, with low pairwise $F_{ST} \approx 0.005$–0.008, between Middle/Late Neolithic and Neolithic western mainland European populations (Fig. 3). Pairwise outgroup-$f_3$ analysis shows a similar pattern, with the highest values of $f_3$ (i.e., most shared drift) being with Western European Neolithic and Copper Age populations (Fig. 3), gradually dropping off for populations more distant in time or space (Supp. Fig. 10).

Ancient Sardinian individuals are shifted towards WHG individuals in the top two PCs relative to early Neolithic Anatolians (Fig. 2). Analysis using qpAdm shows that a two-way admixture model between WHG and Neolithic Anatolian populations is consistent with our data (e.g., $p = 0.376$ for Sar-MN, Table 1), similar to other western European populations of the early Neolithic (Supp. Table 1). The method estimates ancient Sardinian individuals harbor HG ancestry ($\approx 17 \pm 2\%$) that is higher than early Neolithic mainland populations (including Iberia, $8.7 \pm 1.1\%$), but lower than Copper Age Iberians ($25.1 \pm 0.9\%$) and about the same as Southern French Middle-Neolithic individuals ($21.3 \pm 1.5\%$, Table 1, Supp. Fig. 13, $\pm$ denotes plus and minus one standard error).

In explicit models of continuity (using qpAdm, see Methods) the southern French Neolithic individuals (France-N) are consistent with being a single source for Middle/Late Neolithic Sardinia ($p = 0.38$ to reject the model of one population being the direct source of the other); followed by other western populations high in EEF ancestry, though with poor fit (qpAdm $p$-values $< 10^{-5}$, Supp. Table 2). France-N may result in improved fits as it is a better match for the WHG and EEF proportions seen in Middle/Late Neolithic Sardinia (Supp. Table 1). As we discuss below, caution is

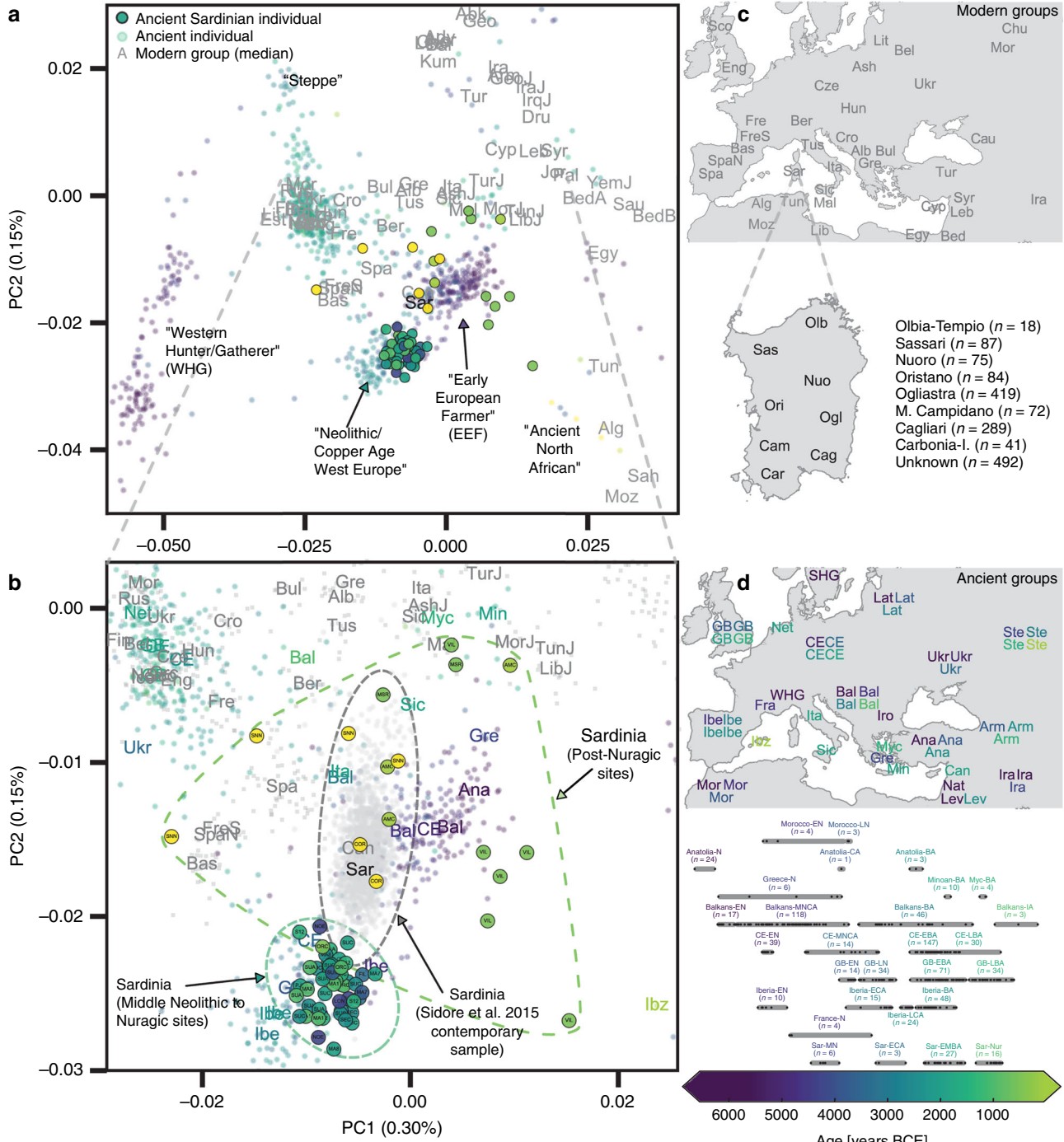

**Fig. 2 Principal components analysis based on the Human Origins dataset. a** Projection of ancient individuals' genotypes onto principal component axes defined by modern Western Eurasians and North Africans (gray labels, see panel (**c**) for legend for all abbreviations but 'Can', for Canary Islands). **b** Zoom into the region most relevant for Sardinia. Each projected ancient individual is displayed as a transparent colored point in panel (**a**) and (**b**), with the color determined by the age of each sample (see panel (**d**) for legend). In panel (**b**), median PC1 and PC2 values for each population are represented by three-letter abbreviations, with black or gray font for moderns and a color-coded font based on the mean age for ancient populations. Ancient Sardinian individuals are plotted as circles with edges, color-coded by age, and with the first three letters of their sample ID (which typically indicates the archeological site). Modern individuals from the Sidore et al. sample of Sardinia are represented with gray circles and modern individuals from the reference panel with gray squares. See Fig. 5 for a zoomed in representation with detailed province labels for Sardinian individuals. The full set of labels and abbreviations are described in Supp. Data 1E, F. **c** Geographic legend of present-day individuals from the Human Origins and our Sardinian reference dataset. **d** Timeline of selected ancient groups. Note: The same geographic abbreviation can appear multiple times with different colors to represent groups with different median ages.

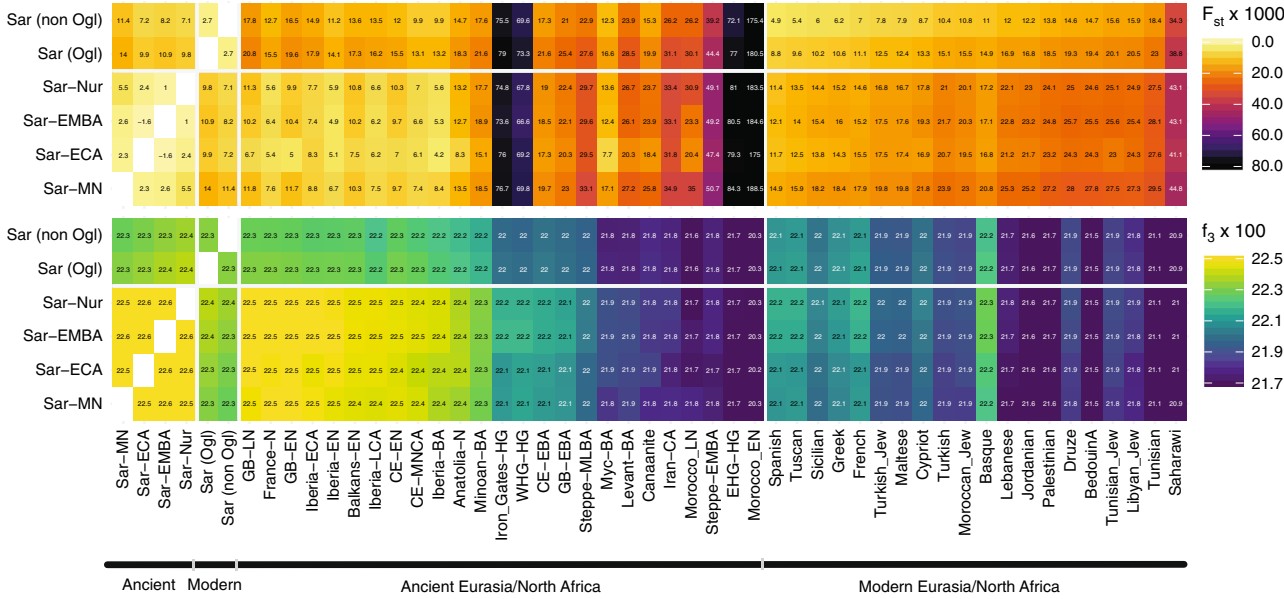

**Fig. 3 Genetic similarity matrices.** We calculated $F_{ST}$ (upper panel) and outgroup-$f_3$ (lower panel) of ancient Sardinian (Middle/Late Neolithic to Nuragic periods) and modern Sardinian individuals (grouped into within and outside the Ogliastra region) with each other (left), various ancient (middle), and modern populations (right) of interest. The full sharing matrices can be found in Supp. Figs. 10/11, where we also include post-Nuragic sites. For ancient groups, sample sizes and time-spans are provided in Fig 1d. The full set of labels and abbreviations are described in Supp. Data 1E, F.

**Table 1 Results from fitting models of admixture with qpAdm for Middle Neolithic to Nuragic period.**

|   | Target | Proxy source populations | | | p-value | Admixture fractions | | | Standard error | | |
|---|---|---|---|---|---|---|---|---|---|---|---|
|   |   | a | b | c |   | a | b | c | a | b | c |
| A | Sar-MN | WHG | Anatolia-N | – | 0.376 | 0.177 | 0.823 | – | 0.014 | 0.014 | – |
|   | Sar-ECA | WHG | Anatolia-N | – | 0.268 | 0.161 | 0.839 | – | 0.020 | 0.020 | – |
|   | Sar-EMBA | WHG | Anatolia-N | – | 0.049 | 0.161 | 0.839 | – | 0.007 | 0.007 | – |
|   | Sar-Nur | WHG | Anatolia-N | – | 0.134 | 0.163 | 0.837 | – | 0.009 | 0.009 | – |
| B | Sar-MN | WHG | Anatolia-N | Steppe | 0.265 | 0.177 | 0.823 | 0.000 | 0.016 | 0.023 | 0.026 |
|   | Sar-ECA | WHG | Anatolia-N | Steppe | 0.18 | 0.164 | 0.836 | 0.000 | 0.023 | 0.032 | 0.036 |
|   | Sar-EMBA | WHG | Anatolia-N | Steppe | 0.032 | 0.162 | 0.838 | 0.000 | 0.009 | 0.012 | 0.013 |
|   | Sar-Nur | WHG | Anatolia-N | Steppe | 0.089 | 0.163 | 0.837 | 0.000 | 0.010 | 0.014 | 0.016 |
| C | France-N | WHG | Anatolia-N | Steppe | 0.093 | 0.213 | 0.787 | 0.000 | 0.018 | 0.023 | 0.027 |
|   | Iberia-EN | WHG | Anatolia-N | Steppe | 0.243 | 0.087 | 0.913 | 0.000 | 0.012 | 0.017 | 0.019 |
|   | Iberia-LCA | WHG | Anatolia-N | Steppe | 0.045 | 0.251 | 0.749 | 0.000 | 0.012 | 0.015 | 0.018 |
|   | Iberia-BA | WHG | Anatolia-N | Steppe | $6.0 \times 10^{-3}$ | 0.239 | 0.689 | 0.072 | 0.010 | 0.014 | 0.016 |
|   | CE-EN | WHG | Anatolia-N | Steppe | 0.656 | 0.046 | 0.954 | 0.000 | 0.007 | 0.010 | 0.012 |
|   | CE-LBA | WHG | Anatolia-N | Steppe | 0.105 | 0.128 | 0.403 | 0.468 | 0.008 | 0.011 | 0.013 |

(A) Two-way models of admixture for ancient Sardinia using Western Hunter-Gatherer (WHG) and Neolithic Anatolia (Anatolia-N) individuals as proxy sources. (B) Three-way models of admixture for ancient Sardinia using Western Hunter-Gatherer (WHG), Neolithic Anatolia (Anatolia-N), and Early Middle Bronze Age Steppe (Steppe-EMBA, abbreviated Steppe in table) individuals as proxy sources. (C) Three-way models for select comparison populations on the European mainland. Full results are reported in Supp. Info. 4.

necessary as there is a lack of aDNA from other relevant populations of the same period (such as mainland Italian Neolithic cultures and neighboring islands).

For our sample from the Middle Neolithic through the Nuragic ($n = 52$ individuals), we were able to infer mtDNA haplotypes for each individual and Y haplotypes for 30 out of 34 males. The mtDNA haplotypes belong to macro-haplogroups HV ($n = 20$), JT ($n = 19$), U ($n = 12$), and X ($n = 1$), a composition broadly similar to other European Neolithic populations. For Y haplotypes, we found at least one carrier for each of three major Sardinia-specific Y founder clades (within the haplogroups I2-M26, G2-L91, and R1b-V88) that were identified previously based on modern Sardinian data[51]. More than half of the 31 identified Y haplogroups were R1b-V88 or I2-M223 ($n = 11$ and 8,

respectively, Supp. Fig. 6, Supp. Data 1B), both of which are also prevalent in Neolithic Iberians[14]. Compared with most other ancient populations in our reference dataset, the frequency of R1b-V88 (Supp. Note 3, Supp. Fig. 6) is relatively high, but as we observed clustering of Y haplogroups by sample location (Supp. Data 1B) caution should be exercised with interpreting our results as estimates for island-wide Y haplogroup frequencies. The oldest individuals in our reference data carrying R1b-V88 or I2-M223 were Balkan hunter-gatherer and Neolithic individuals, and both haplogroups later appear also in western Neolithic populations (Supp. Figs. 7–9).

**Continuity from the Middle Neolithic through the Nuragic.** We found several lines of evidence supporting genetic continuity

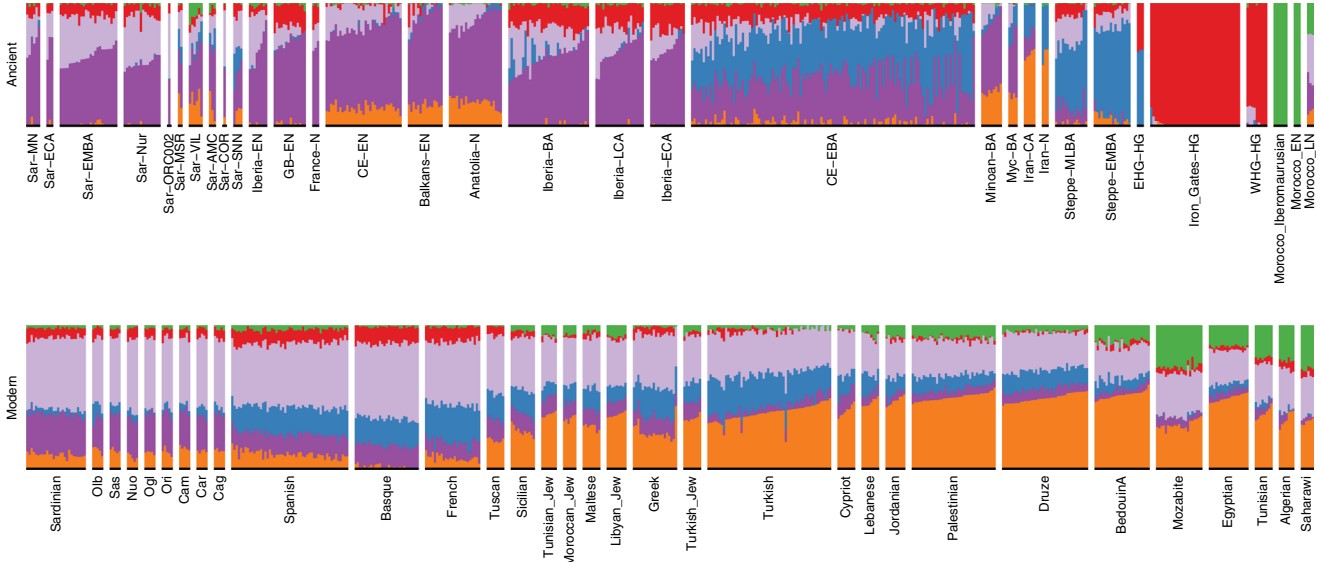

**Fig. 4 Admixture coefficients estimated by ADMIXTURE (K = 6).** Each stacked bar represents one individual and color fractions depict the fraction of the given individual's ancestry coming from a given "cluster". For K = 6 (depicted here), Sardinian individuals up until the Nuragic share similar admixture proportions as other western European Neolithic individuals. Present-day as well as most post-Nuragic ancient Sardinian individuals have elevated Steppe-like ancestry (blue), and an additional ancestry component prevalent in Near Eastern/Levant populations (orange). An ancient North-African component (green) appears at low fraction in many present-day Mediterranean populations, and somewhat stronger in samples from the Sardinian Punic site Villamar. ADMIXTURE results for all K = 2, …, 11 are depicted in the supplement (Supp. Fig. 19).

from the Sardinian Middle Neolithic into Bronze Age and Nuragic times. Importantly, we observed low genetic differentiation between ancient Sardinian individuals from various time periods ($F_{ST} = 0.0055 \pm 0.0014$ between Middle/Late Neolithic and late Bronze Age, Fig. 3). Furthermore, we did not observe temporal substructure within the ancient Sardinian individuals in the top two PCs—they form a coherent cluster (Fig. 2). In stark contrast, ancient individuals from mainland regions such as central Europe show large movements over the first two PCs from the Neolithic to the Bronze Age, and also have higher pairwise differentiation (e.g., $F_{ST} = 0.0200 \pm 0.0004$ between Neolithic and Bronze Age individuals from central Europe, Supp. Fig. 11). A qpAdm analysis cannot reject a model of Middle/Late Neolithic Sardinian individuals being a direct predecessor of Nuragic Sardinian individuals ($p = 0.15$, Supp. Table 2, also see results for $f_4$ statistics, Supp. Data 2). Our qpAdm analysis further shows that the WHG ancestry proportion, in a model of admixture with Neolithic Anatolia, remains stable at $17 \pm 2\%$ through the Nuragic period (Table 1A). When using a three-way admixture model, we do not detect significant Steppe ancestry in any ancient Sardinian group from the Middle/Late Neolithic to the Nuragic, as is inferred, for example, in later Bronze Age Iberians (Table 1B, Supp. Fig. 13). Finally, in a five-way model with Iran Neolithic and Moroccan Neolithic samples added as sources, neither source is inferred to contribute ancestry during the Middle Neolithic to Nuragic (point estimates are statistically indistinguishable from zero, Supp. Fig. 14).

**From the Nuragic period to present-day Sardinia: signatures of admixture.** We found multiple lines of evidence for gene flow into Sardinia after the Nuragic period. The present-day Sardinian individuals from the Sidore et al. sample are shifted from the Nuragic period ancients on the western Eurasian/North-African PCA (Fig. 2). Using a "shrinkage" correction method for the projection is key for detecting this shift (see Supp. Fig. 23 for an evaluation of different PCA projection techniques). In the ADMIXTURE results (Fig. 4), present-day Sardinian individuals

carry a modest "Steppe-like" ancestry component (but generally less than continental present-day European populations), and an appreciable "eastern Mediterranean" ancestry component (also inferred at a high fraction in other present-day Mediterranean populations, such as Sicily and Greece) relative to Nuragic period and earlier Sardinian individuals.

To further refine this recent admixture signal, we considered two-way, three-way, and four-way models of admixture with qpAdm (Table 2, Supp. Figs. 15–18, Supp. Tables 3–5). We find three-way models fit well ($p > 0.01$) that contain admixture between Nuragic Sardinia, one northern Mediterranean source (e.g., individuals with group labels Lombardy, Tuscan, French, Basque, Spanish) and one eastern Mediterranean source (e.g., individuals with group labels Turkish-Jew, Libyan-Jew, Maltese, Tunisian-Jew, Moroccan-Jew, Lebanese, Druze, Cypriot, Jordanian, Palestinian) (Table 2C, D). Maltese and Sicilian individuals can provide two-way model fits (Table 2B), but appear to reflect a mixture of N. Mediterranean and E. Mediterranean ancestries, and as such they can serve as single-source proxies in two-way admixture models with Nuragic Sardinia. For four-way models including N. African ancestry, the inferences of N. African ancestry are negligible (though as we show below, forms of N. African ancestry were already likely present in the eastern Mediterranean components).

Because of limited sample sizes and ancestral source misspecification, caution is warranted when interpreting inferred admixture fractions; however, the results indicate that complex post-Nuragic gene flow has likely played a role in the population genetic history of Sardinia.

**Refined signatures of post-Nuragic admixture and heterogeneity.** To more directly evaluate the models of post-Nuragic admixture, we obtained aDNA from 17 individuals sampled from post-Nuragic sites. The post-Nuragic individuals spread across a wide range of the PCA, and many shift towards the "eastern" and "northern" Mediterranean sources posited above (Fig. 2). We confidently reject qpAdm models of continuity from the Nuragic

**Table 2 Results from fitting models of admixture with qpAdm and Sardinian aDNA as sources.**

| | Target | Proxy source populations | | | p-value | Admixture fractions | | | Standard error | | |
|---|---|---|---|---|---|---|---|---|---|---|---|
| | | a | b | c | | a | b | c | a | b | c |
| A | Sar-ECA | Sar-MN | – | – | 0.175 | – | – | – | – | – | – |
| | Sar-EMBA | Sar-ECA | – | – | 0.769 | – | – | – | – | – | – |
| | Sar-Nur | Sar-EMBA | – | – | 0.765 | – | – | – | – | – | – |
| | Cagliari | Sar-Nur | – | – | $<10^{-30}$ | – | – | – | – | – | – |
| B | Cagliari | Sar-Nur | Sicilian | – | 0.011 | 0.545 | 0.455 | – | 0.021 | 0.021 | – |
| | Cagliari | Sar-Nur | Maltese | – | 0.011 | 0.573 | 0.427 | – | 0.019 | 0.019 | – |
| | Cagliari | Sar-Nur | Turkish | – | $2.4 \times 10^{-3}$ | 0.699 | 0.301 | – | 0.014 | 0.014 | – |
| | Cagliari | Sar-Nur | Tuscan | – | $2.3 \times 10^{-4}$ | 0.522 | 0.478 | – | 0.024 | 0.024 | – |
| C | Cagliari | Sar-Nur | N Mediterranean | Turkish-Jew | 0.168 | 0.512 | 0.237 | 0.251 | 0.021 | 0.042 | 0.034 |
| | Cagliari | Sar-Nur | N Mediterranean | Libyan-Jew | 0.046 | 0.515 | 0.299 | 0.186 | 0.023 | 0.038 | 0.028 |
| | Cagliari | Sar-Nur | N Mediterranean | Tunisian-Jew | 0.037 | 0.498 | 0.312 | 0.191 | 0.022 | 0.035 | 0.028 |
| | Cagliari | Sar-Nur | N Mediterranean | Druze | 0.031 | 0.542 | 0.287 | 0.170 | 0.022 | 0.037 | 0.024 |
| | Cagliari | Sar-Nur | N Mediterranean | Moroccan-Jew | 0.026 | 0.519 | 0.275 | 0.206 | 0.022 | 0.042 | 0.032 |
| | Cagliari | Sar-Nur | N Mediterranean | Cypriot | 0.026 | 0.520 | 0.297 | 0.184 | 0.021 | 0.035 | 0.026 |
| | Cagliari | Sar-Nur | N Mediterranean | Maltese | 0.025 | 0.541 | 0.132 | 0.327 | 0.025 | 0.069 | 0.058 |
| | Cagliari | Sar-Nur | N Mediterranean | Lebanese | 0.023 | 0.550 | 0.299 | 0.151 | 0.024 | 0.038 | 0.023 |
| | Cagliari | Sar-Nur | N Mediterranean | Sicilian | 0.021 | 0.526 | 0.113 | 0.361 | 0.022 | 0.064 | 0.058 |
| | Cagliari | Sar-Nur | N Mediterranean | Jordanian | $10.0 \times 10^{-3}$ | 0.542 | 0.319 | 0.138 | 0.023 | 0.035 | 0.021 |
| | Cagliari | Sar-Nur | N Mediterranean | Greek | $8.7 \times 10^{-3}$ | 0.551 | 0.000 | 0.449 | 0.037 | 0.217 | 0.191 |
| | Cagliari | Sar-Nur | N Mediterranean | Palestinian | $7.0 \times 10^{-3}$ | 0.542 | 0.331 | 0.127 | 0.024 | 0.035 | 0.019 |
| | Cagliari | Sar-Nur | N Mediterranean | Turkish | $6.3 \times 10^{-3}$ | 0.637 | 0.136 | 0.228 | 0.033 | 0.067 | 0.039 |
| | Cagliari | Sar-Nur | N Mediterranean | BedouinA | $2.0 \times 10^{-3}$ | 0.540 | 0.351 | 0.109 | 0.024 | 0.033 | 0.017 |
| | Cagliari | Sar-Nur | N Mediterranean | Egyptian | $1.4 \times 10^{-4}$ | 0.529 | 0.389 | 0.082 | 0.025 | 0.031 | 0.014 |
| | Cagliari | Sar-Nur | N Mediterranean | Tunisian | $3.7 \times 10^{-5}$ | 0.524 | 0.404 | 0.072 | 0.025 | 0.031 | 0.014 |
| D | Cagliari | Sar-Nur | E Mediterranean | Lombardy | 0.168 | 0.512 | 0.251 | 0.237 | 0.021 | 0.034 | 0.042 |
| | Cagliari | Sar-Nur | E Mediterranean | Tuscan | 0.09 | 0.527 | 0.198 | 0.275 | 0.020 | 0.047 | 0.053 |
| | Cagliari | Sar-Nur | E Mediterranean | Greek | 0.079 | 0.548 | 0.151 | 0.302 | 0.018 | 0.054 | 0.057 |
| | Cagliari | Sar-Nur | E Mediterranean | French | 0.05 | 0.560 | 0.324 | 0.116 | 0.019 | 0.027 | 0.023 |
| | Cagliari | Sar-Nur | E Mediterranean | Basque | 0.034 | 0.533 | 0.340 | 0.128 | 0.021 | 0.025 | 0.025 |
| | Cagliari | Sar-Nur | E Mediterranean | Spanish | 0.023 | 0.540 | 0.309 | 0.151 | 0.020 | 0.029 | 0.030 |
| | Cagliari | Sar-Nur | E Mediterranean | Sicilian | 0.013 | 0.544 | 0.000 | 0.456 | 0.029 | 0.226 | 0.245 |
| | Cagliari | Sar-Nur | E Mediterranean | Maltese | 0.012 | 0.572 | 0.000 | 0.428 | 0.026 | 0.261 | 0.266 |
| | Cagliari | Sar-Nur | E Mediterranean | Turkish | $1.2 \times 10^{-3}$ | 0.700 | 0.000 | 0.300 | 0.058 | 0.190 | 0.135 |
| | Cagliari | Sar-Nur | E Mediterranean | Cypriot | $3.8 \times 10^{-5}$ | 0.587 | 0.413 | 0.000 | 0.047 | 0.310 | 0.275 |
| E | Sar-VIL | Sar-Nur | – | – | $<10^{-30}$ | – | – | – | – | – | – |
| | Sar-MSR | Sar-VIL | – | – | $1.8 \times 10^{-6}$ | – | – | – | – | – | – |
| | Sar-AMC | Sar-MSR | – | – | 0.203 | – | – | – | – | – | – |
| | Sar-SNN | Sar-MSR | – | – | 0.037 | – | – | – | – | – | – |
| | Sar-COR | Sar-AMC | – | – | 0.014 | – | – | – | – | – | – |
| | Sar-SNN | Sar-AMC | – | – | 0.124 | – | – | – | – | – | – |
| F | Cagliari | Sar-VIL | – | – | $9.1 \times 10^{-12}$ | – | – | – | – | – | – |
| | Cagliari | Sar-MSR | – | – | 0.078 | – | – | – | – | – | – |
| | Cagliari | Sar-AMC | – | – | 0.012 | – | – | – | – | – | – |
| | Cagliari | Sar-COR | – | – | 0.16 | – | – | – | – | – | – |
| | Cagliari | Sar-SNN | – | – | 0.037 | – | – | – | – | – | – |
| | Ogliastra | Sar-VIL | – | – | $8.6 \times 10^{-14}$ | – | – | – | – | – | – |
| | Ogliastra | Sar-MSR | – | – | 0.044 | – | – | – | – | – | – |
| | Ogliastra | Sar-AMC | – | – | $2.2 \times 10^{-3}$ | – | – | – | – | – | – |
| | Ogliastra | Sar-COR | – | – | 0.261 | – | – | – | – | – | – |
| | Ogliastra | Sar-SNN | – | – | 0.016 | – | – | – | – | – | – |

(A) Single-source models to assess continuity of each Sardinian period with the previous one (see main text for guide to abbreviations). (B) Results of two-way models of admixture for Cagliari (a representative present-day sample). (C) Results of three-way models showing multiple eastern Mediterranean populations that can produce viable models (Results shown with individuals from Lombardy [Bergamo in the Human Origins array (HOA) dataset, see Methods] as one of several possible proxies for north Mediterranean ancestry, see part C). (C) Results of three-way models showing multiple north Mediterranean populations that can produce viable models (Results shown with Jewish individuals from Turkey ['Turkish-Jew' in the dataset] used as one of several possible proxies for east Mediterranean ancestry, see part B). (E) Results of single-source models to assess continuity among post-Nuragic sites. (F) Results of single-source models to assess continuity between the Medieval period samples and present-day samples (Cagliari and Ogliastra taken as representatives). Full results are reported in Supp. Info. 4.

period for all of these post-Nuragic samples, apart from a sample from S'Orcu 'e Tueri (ORC002, Table 2E, Supp. Table 6). The ADMIXTURE results concur, most post-Nuragic individuals show the presence of novel ancestry components not inferred in any of the more ancient individuals (Fig. 4).

Consistent with an influx of novel ancestry, we observed that haplogroup diversity increases after the Nuragic period. In particular, we identified one carrier of the mtDNA haplogroup L2a at both the Punic Villamar site and the Roman Monte Carru site. At present, this mtDNA haplogroup is common across Africa, but so far undetected in samples from Sardinia[36]. We also found several Y haplogroups absent in our Neolithic trough the Nuragic period sample (Supp. Fig. 6). R1b-M269, at about 15% within modern Sardinian males[51], appears in one Punic (VIL011) and two Medieval individuals (SNN002 and SNN004). We also observed J1-L862 in one individual from a Punic site (VIL007) and E1b-L618 in one medieval individual (SNN001). Notably, J1-L862 first appears in Levantine Bronze Age individuals within the ancient reference dataset and is at about 5% frequency in Sardinia today.

We used individual-level qpAdm models to further investigate the presence of these new ancestries (Supp. Data 3). In addition to the original Neolithic Anatolian (Anatolia-N) and Hunter Gatherer (WHG) sources that were sufficient to model ancient Sardinians through the Nuragic period, we fit models with representatives of Steppe (Steppe-EMBA), Neolithic Iranian (Iran-N), and Neolithic North-African (Morocco-EN) ancestry as sources. We observe the presence of the Steppe-EMBA (point estimates ranging 0–20%) and Iran-N components (point estimates ranging 0–25%) in many of the post-Nuragic individuals (Supp. Fig. 14). All six individuals from the Punic Villamar site were inferred to have substantial levels of ancient North-African ancestry (point estimates ranging 20–35%, Supp. Fig. 14, also see ADMIXTURE and PCA results, Figs. 2 and 4). When fit with the same five-way admixture model, present-day Sardinians have a small but detectable level of North-African ancestry (Supp. Fig. 14, also see ADMIXTURE analysis, Fig. 4).

Models with direct continuity from Villamar to the present are rejected (Table 2F, Supp. Table 6). In contrast, nearly all the other post-Nuragic sites produce viable models as single sources for the

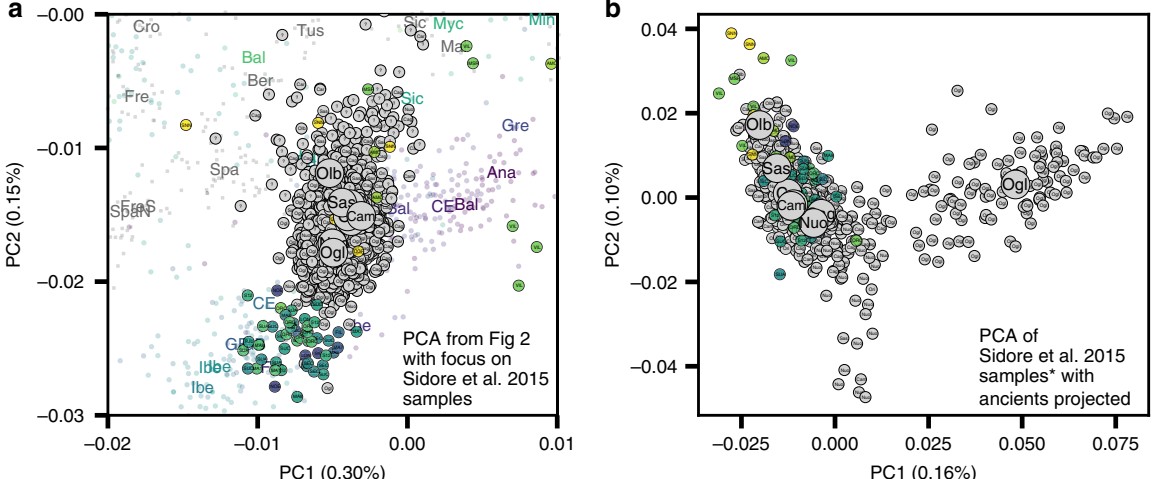

**Fig. 5 Present-day genetic structure in Sardinia reanalyzed with aDNA. a** Scatter plot of the first two principal components from Fig. 2a with a zoom-in on present-day Sardinia diversity in our sample[33]. Median PC values for each Sardinian region are depicted as large circles. **b** PCA results based on present-day Sardinian individuals, subsampling Cagliari and Ogliastra to 100 individuals to avoid effects of unbalanced sampling. In both panels, each individual is labeled with an abbreviation that denotes the source location if at least three grandparents were born in the same geographical location ("small" three-letter abbreviations) or if grand-parental ancestry is missing with question mark. We also projected each ancient Sardinian individual on to the top two PCs (points color-coded by age, see Fig. 1 for the color scale).

modern Sardinians (e.g., Sar-COR qpAdm p-values of 0.16 and 0.261 for Cagliari and Ogliastra, respectively; Sar-SNN qpAdm p-values of 0.037 and 0.016, similarly Table 2F, Supp. Table 6). We found some evidence of substructure: Sar-ORC002 (from an interior site) is more consistent with being a single source for Ogliastra than Cagliari, whereas Sar-AMC shows an opposite pattern (Supp. Table 6).

We also carried out three-way admixture models for each post-Nuragic Sardinian individual using the Nuragic sample as a source or outgroup, and potential sources from various ancient samples that are representative of different regions of the Mediterranean. We found a range of models can be fit for each individual (Supp. Tables 7–8). For the models with Nuragic as a source, by varying the proxy populations, one can obtain fitted models that vary widely in the inferred Nuragic component (e.g., individual COR002 has a range from 4.4 to 87.8% across various fitted models; similarly, individual AMC001, with North-African mtDNA haplogroup U6a, had a range form 0.2 to 43.1%, see Supp. Tables 7–8). The ORC002 sample had the strongest evidence of Nuragic ancestry (range from 62.8 to 96.3%, see Supp. Tables 7–8). Further, the VIL, MSR, and AMC individuals can be modeled with Nuragic Sardinian individuals included as a source or as an outgroup, while the two COR and ORC002 individuals can only be modeled with Nuragic individuals included as a source. One individual from the medieval period San Nicola Necropoli (SNN001) was distinct in that we found their ancestry can be modeled in a single-source model as descendant of a population represented by present-day Basque individuals (Supp. Table 8). When we apply the same approach to present-day Sardinian individuals, we find models with the Nuragic sample as an outgroup fail in most cases (Supp. Table 9). For models that include Nuragic as a possible source, each present-day individual is consistent with a wide range of Nuragic ancestry. The models with the largest p-values return fractions of Nuragic ancestry that are close to, or higher than 50% (Supp. Table 9), similar to observed in our population-level modeling (Table 2).

**Fine-scale structure in contemporary Sardinia.** Finally, we assessed our results in the context of spatial substructure within

modern Sardinia, as previous studies have suggested elevated levels of WHG and EEF ancestry in Ogliastra[32].

In the PCA of modern west Eurasian and North-African variation, the ancient Sardinian individuals are placed closest to individuals from Ogliastra and Nuoro (see Figs. 2 and 5a). At the same time, in a PCA of just the modern Sardinian sample, the ancient individuals project furthest from Ogliastra (Fig. 5b). Interestingly, individual ORC002, dating from the Punic period and from a site in Ogliastra, projects towards Ogliastra individuals relative to other ancient individuals.

Further, in the broad PCA results, the median of the province of Olbia-Tempio (northeast Sardinia) is shifted towards mainland populations of southern Europe, and the median for Campidano (southwest Sardinia) shows a slight displacement towards the eastern Mediterranean (Fig. 5a). A three-way admixture model fit with qpAdm suggests differential degrees of admixture, with the highest eastern Mediterranean ancestry in the southwest (Carbonia, Campidano) and the highest northern Mediterranean ancestry in the northeast of the island (Olbia, Sassari, Supp. Fig. 17). These observations of substructure among contemporary Sardinian individuals contrast our results from the Nuragic and earlier, which forms a relatively tight cluster on the broad PCA (Fig. 2) and for which the top PCs do not show any significant correlations with latitude, longitude, or regional geographic labels after correcting for multiple testing (Supp. Figs. 24–33).

## Discussion

Our analysis of genome-wide data from 70 ancient Sardinian individuals has generated insights regarding the population history of Sardinia and the Mediterranean. First, our analysis provides more refined DNA-based support for the Middle Neolithic of Sardinia being related to the early Neolithic peoples of the Mediterranean coast of Europe. Middle/Late Neolithic Sardinian individuals fit well as a two-way admixture between mainland EEF and WHG sources, similar to other EEF populations of the western Mediterranean. Further, we detected Y haplogroups R1b-V88 and I2-M223 in the majority of the early Sardinian males. Both haplogroups appear earliest in the Balkans among Meso-lithic hunter-gatherers and then Neolithic groups[9] and later in EEF Iberians[14], in which they make up the majority of Y

haplogroups, but have not been detected in Neolithic Anatolians or more western WHG individuals. These results are plausible outcomes of substantial gene flow from Neolithic populations that spread westward along the Mediterranean coast of southern Europe around 5500 BCE (a "Cardial/Impressed" ware expansion, see Introduction). We note that we lack autosomal aDNA from earlier than the Middle Neolithic in Sardinia and from key mainland locations such as Italy, which leaves some uncertainty about timing and the relative influence of gene flow from the Italian mainland versus from the north or west. The inferred WHG admixture fraction of Middle Neolithic Sardinians was higher than that of early mainland EEF populations, which could suggest a time lag of the influx into Sardinia (as HG ancestry increased through time on the mainland) but also could result from a pulse of initial local admixture or continued gene flow with the mainland. Genome-wide data from Mesolithic and early Neolithic individuals from Sardinia and potential source populations will help settle these questions.

From the Middle Neolithic onward until the beginning of the first millennium BC, we do not find evidence for gene flow from distinct ancestries into Sardinia. That stability contrasts with many other parts of Europe which had experienced substantial gene flow from central Eurasian Steppe ancestry starting about 3000 BCE[11,12] and also with many earlier Neolithic and Copper age populations across mainland Europe, where local admixture increased WHG ancestry substantially over time[10]. We observed remarkable constancy of WHG ancestry (close to 17%) from the Middle Neolithic to the Nuragic period. While we cannot exclude influx from genetically similar populations (e.g., early Iberian Bell Beakers), the absence of Steppe ancestry suggests genetic isolation from many Bronze Age mainland populations—including later Iberian Bell Beakers[13]. As further support, the Y haplogroup R1b-M269, the most frequent present-day western European haplogroup and associated with expansions that brought Steppe ancestry into Britain[13] and Iberia[14] about 2500–2000 BCE, remains absent in our Sardinian sample through the Nuragic period (1200–1000 BCE). Larger sample sizes from Sardinia and alternate source populations may discover more subtle forms of admixture, but the evidence appears strong that Sardinia was isolated from major mainland Bronze Age gene flow events through to the local Nuragic period. As the archeological record shows that Sardinia was part of a broad Mediterranean trade network during this period[19], such trade was either not coupled with gene flow or was only among proximal populations of similar genetic ancestry. In particular, we find that the Nuragic period is not marked by shifts in ancestry, arguing against hypotheses that the design of the Nuragic stone towers was brought with an influx of people from eastern sources such as Mycenaeans.

Following the Nuragic period, we found evidence of gene flow with both northern and eastern Mediterranean sources. We observed eastern Mediterranean ancestry appearing first in two Phoenician-Punic sites (Monte Sirai, Villamar). The northern Mediterranean ancestry became prevalent later, exemplified most clearly by individuals from a north-western Medieval site (San Nicola Necropoli). Many of the post-Nuragic individuals could be modeled as direct immigrants or offspring from new arrivals to Sardinia, while others had higher fractions of local Nuragic ancestry (Corona Moltana, ORC002). Substantial uncertainty exists here as the low differentiation among plausible source populations makes it challenging to exclude alternate models, especially when using individual-level analysis. Overall though, we find support for increased variation in ancestry after the Nuragic period, and this echoes other recent aDNA studies in the Mediterranean that have observed fine-scale local heterogeneity in the Iron Age and later[14,52–54].

In addition, we found present-day Sardinian individuals sit within the broad range of ancestry observed in our ancient samples. A similar pattern is seen in Iberia[14] and central Italy[54], where variation in individual ancestry increased markedly in the Iron Age, and later decreased until present-day. In terms of the fine-scale structure within Sardinia, we note the median position of modern individuals from the central regions of Ogliastra and Nuoro on the main PCA (Fig. 5a) are less shifted towards novel sources of post-Nuragic admixture, which reinforces a previous result that Ogliastra shows higher levels of EEF and HG ancestry than other regions[32]. At the same time, in the PCA of within Sardinia variation (Fig. 5b), differentiation of Ogliastra from other regions and other ancient individuals is apparent, likely reflecting a recent history of isolation and drift. The northern provinces of Olbia-Tempio, and to a lesser degree Sassari, appear to have received more northern Mediterranean immigration after the Bronze Age; while the southwestern provinces of Campidano and Carbonia carry more eastern Mediterranean ancestry. Both of these results align with known history: the major Phoenician and Punic settlements in the first millennium BCE were situated principally along the south and west coasts, and Corsican shepherds, speaking an Italian-Corsican dialect (Gallurese), immigrated to the northeastern part of Sardinia[55].

Our inference of gene flow after the second millennium BCE seems to contradict previous models emphasizing Sardinian isolation[12]. These models were supported by admixture tests that failed to detect substantial admixture[32], likely because of substantial drift and a lack of a suitable proxy for the Nuragic Sardinian ancestry component. However, compared with other European populations[50,56], we confirm Sardinia experienced relative genetic isolation through the Bronze Age/Nuragic period. In addition, we find that subsequent admixture appears to derive mainly from Mediterranean sources that have relatively little Steppe ancestry. Consequently, present-day Sardinian individuals have retained an exceptionally high degree of EEF ancestry and so they still cluster with several mainland European Copper Age individuals such as Ötzi[2], even as they are shifted from ancient Sardinian individuals of a similar time period (Fig. 2).

The Basque people, another population high in EEF ancestry, were previously suggested to share a genetic connection with modern Sardinian individuals[32,57]. We observed a similar signal, with modern Basque having, of all modern samples, the largest pairwise outgroup-$f_3$ with most ancient and modern Sardinian groups (Fig. 3). While both populations have received some immigration, seemingly from different sources (e.g., Fig. 4, ref. [14]), our results support that the shared EEF ancestry component could explain their genetic affinity despite their geographic separation.

Beyond our focal interest in Sardinia, the results from individuals from the Phoenician-Punic sites Monte Sirai and Villamar shed some light on the ancestry of a historically impactful Mediterranean population. Notably, they show strong genetic relationships to ancient North-African and eastern Mediterranean sources. These results mirror other emerging ancient DNA studies[37,58], and are not unexpected given that the Punic center of Carthage on the North-African coast itself has roots in the eastern Mediterranean. Interestingly, the Monte Sirai individuals, predating the Villamar individuals by several centuries, show less North-African ancestry. This could be because they harbor earlier Phoenician ancestry and North-African admixture may have been unique to the later Punic context, or because they were individuals from a different ancestral background altogether. Estimated North-African admixture fractions were much lower in later ancient individuals and present-day Sardinian individuals, in line with previous studies that have observed small but significant

African admixture in several present-day South European populations, including Sardinia[32,59,60].

As ancient DNA studies grow, a key challenge will be fine-scale sampling to aid the interpretation of shifts in ancestry. Our sample from Sardinia's post-Nuragic period highlights the complexity, as we simultaneously observe examples of individuals that appear as novel immigrant ancestries (e.g., from Villamar and San Nicola) and of individuals that look more contiguous to the past and to the present (e.g., the two Corona Moltana siblings, the ORC002 individual, several of the Alghero Monte Carru individuals). This variation is likely driven by differential patterns of contact—as might arise between coastal versus interior villages, central trading centers versus remote agricultural sites, or even between neighborhoods and social strata in the same village. We also note that modern populations are collected with different biases than ancient individuals (e.g., the sub-populations sampled by medical genetics projects[33] versus the sub-populations that are accessible at archeological sites). As such, caution should be exercised when generalizing from the sparse sampling typical for many aDNA studies, including this one.

With these caveats in mind, we find that genome-wide ancient DNA provides unique insights into the population history of Sardinia. Our results are consistent with gene flow being minimal or only with genetically similar populations from the Middle Neolithic until the late Bronze Age. In particular, the onset of the Nuragic period was not characterized by influx of a distinct ancestry. The data also link Sardinia from the Iron Age onwards to the broader Mediterranean in what seems to have been a period of new dynamic contact throughout much of the Mediterranean. A parallel study focusing on islands of the western Mediterranean provides generally consistent results and both studies make clear the need to add complexity to simple models of sustained isolation that have dominated the genetic literature on Sardinia[52]. Finally, our results suggest some of the current substructure seen on the island (e.g., Ogliastra) has emerged due to recent genetic drift. Overall, the history of isolation, migration, and genetic drift on the island has given rise to an unique constellation of allele frequencies, and illuminating this history will help future efforts to understand genetic-disease variants prevalent in Sardinia and throughout the Mediterranean, such as those underlying beta-thalassemia and G6PD deficiency.

## Methods

**Archeological sampling**. The archeological samples used in this project derive from several collection avenues. The first was a sampling effort led by co-author Luca Lai, leveraging a broad base of samples from different existing collections in Sardinia, a subset of which were previously used in isotopic analyses to understand dietary composition and change in prehistoric Sardinia[40]. The second was from the Seulo Caves project[41], an on-going project on a series of caves that span the Middle Neolithic to late Bronze Age near the town of Seulo. The project focuses on the diverse forms and uses of caves in the prehistoric culture of Sardinia. The Neolithic individuals from Sassari province as well as the post-Nuragic individuals were collected from several co-authors as indicated in Supplementary Information Section 1. The third was a pair of Neolithic sites Noedalle and S'isterridolzu[42]. The fourth are a collection of post-Nuragic sites spanning from the Phoenician to the Medieval time. All samples were handled in collaboration with local scientists and with the approval of the local Sardinian authorities for the handling of archeological samples (Ministero per i Beni e le Attività Culturali, Direzione Generale per i beni Archeologici, request dated 11 August 2009; Soprintendenza per i Beni Archeologici per le province di Sassari e Nuoro, prot. 12993 dated 20 December 2012; Soprintendenza per i Beni Archeologici per le province di Sassari e Nuoro, prot. 10831 dated 27 October 2014; Soprintendenza per i Beni Archeologici per le province di Sassari e Nuoro, prot. 12278 dated 05 December 2014; Soprintendenza per i Beni Archeologici per le Provincie di Cagliari e Oristano, prot. 62, dated 08 January 2015; Soprintendenza Archeologia, Belle Arti e Paesaggio per le Provincie di Sassari, Olbia-Tempio e Nuoro, prot. 4247 dated 14 March 2017; Soprintendenza per i Beni Archeologici per le Provincie di Sassari e Nuoro, prot. 12930 dated 30 December 2014; Soprintendenza Archeologia, Belle arti e Paesaggio per le Provincie di Sassari e Nuoro, prot. 7378 dated 9 May, 2017; Soprintendenza per i Beni Archeologici per le Provincie di Cagliari e Oristano, prot. 20587, dated 05 October 2017; Soprintendenza Archeologia, Belle Arti e Paesaggio per le Provincie

di Sassari e Nuoro, prot. 15796 dated 25 October, 2017; Soprintendenza Archeologia, Belle Arti e Paesaggio per le Provincie di Sassari e Nuoro, prot. 16258 dated 26 November 2017; Soprintendenza per i Beni Archeologici per le province di Sassari e Nuoro, prot. 5833 dated 16 May 2018; Soprintendenza Archeologia, Belle Arti e Paesaggio per la città metropolitana di Cagliari e le province di Oristano e Sud Sardegna, prot. 30918 dated 10 December 2019). For more detailed description of the sites, see Supplementary Information Section 1.

**Initial sample screening and sequencing**. The ancient DNA (aDNA) workflow was implemented in dedicated facilities at the Palaeogenetic Laboratory of the University of Tübingen and at the Department of Archaeogenetics of the Max Planck Institute for the Science of Human History in Jena. The only exception was for four samples from the Seulo Cave Project which had DNA isolated at the Australian Centre for Ancient DNA and capture and sequencing carried out in the Reich lab at Harvard University. Different skeletal elements were sampled using a dentist drill to generate bone and tooth powder, respectively. DNA was extracted following an established aDNA protocol[61] and then converted into double-stranded libraries retaining[62] or partially reducing[63] the typical aDNA substitution pattern resulting from deaminated cytosines that accumulate towards the molecule's termini. After indexing PCR[62] and differential amplification cycles, the DNA was shotgun sequenced on Illumina platforms. Samples showing sufficient aDNA preservation where captured for mtDNA and ≈1.24 million SNPs across the human genome chosen to intersect with the Affymetrix Human Origins array and Illumina 610-Quad array[47]. The resulting enriched libraries were also sequenced on Illumina machines in single-end or paired-end mode. Sequenced data were pre-processed using the EAGER pipeline[64]. Specifically, DNA adapters were trimmed using AdapterRemoval v2[65] and paired-end sequenced libraries were merged. Sequence alignment to the mtDNA (RSRS) and nuclear (hg19) reference genomes was performed with BWA[66] (parameters –n 0.01, seeding disabled), duplicates were removed with DeDup[64] and a mapping quality filter was applied (MQ≥30). For genetic sexing, we compared relative X and Y-chromosome coverage to the autosomal coverage with a custom script. For males, nuclear contamination levels were estimated based on heterozygosity on the X-chromosome with the software ANGSD[67].

After applying several standard ancient DNA quality control metrics, retaining individuals with endogenous DNA content in shotgun sequencing >0.2%, mtDNA contamination <4% (average 1.6%), and nuclear contamination <6% (average 1.1%), and after inspection of contamination patterns (Supp. Figs. 2–5), we generated genotype calls for downstream population genetic analyses for a set of 70 individuals. To account for sequencing errors we first removed any reads that overlapped a SNP on the capture array with a base quality score <20. We also removed the last 3-bp on both sides of every read to reduce the effect of DNA damage on the resulting genotype calls[68]. We used custom python scripts (https://github.com/mathii/gdc3) to generate pseudo-haploid genotypes by sampling a random read for each SNP on the capture array and setting the genotype to be homozygous for the sampled allele. We then screened for first degree relatives using a pairwise relatedness statistic, and identified one pair of siblings and one parent-offspring pair within our sample (Supp. Fig. 12).

**Processing of mtDNA data**. Data originating from mtDNA capture were processed with schmutzi[69], which jointly estimates mtDNA contamination and reconstructs mtDNA consensus sequences that were assigned to the corresponding mtDNA haplogroups using Haplofind[70] (Supp. Data 1C). The consensus sequences were also compared with rCRS[71] to build a phylogenetic tree of ancient Sardinian mitogenomes (Supp. Data 1D) using a maximum parsimony approach with the software mtPhyl (http://eltsov.org/mtphyl.aspx). We assigned haplogroups following the nomenclature proposed by the PhyloTree database build 17 (http://www.phylotree.org)[72] and for Sardinian-specific haplogroups[36].

**Inference of Y haplogroups**. To determine the haplotype branch of the Y chromosome of male ancient individuals, we analyzed informative SNPs on the Y-haplotype tree. For reference, we used markers from https://isogg.org/tree (Version: 13.238, 2018). We merged this data with our set of calls and identified markers available in both to create groups of equivalent markers for sub-haplogroups. Our targeted sequencing approach yielded read count data for up to 32,681 such Y-linked markers per individual. As the conventions for naming of haplogroups are subject to change, we annotated them in terms of carrying the derived state at a defining SNP. We analyzed the number of derived and ancestral calls for each informative marker for all ancient Sardinian individuals and reanalyzed male ancient West Eurasians in our reference dataset. Refined haplotype calls were based on manual inspection of ancestral and derived read counts per haplogroup, factoring in coverage and error estimates.

**Merging newly generated data with published data**. *Ancient DNA datasets from Western Eurasia and North Africa*: We downloaded and processed BAM files from several ancient datasets from continental Europe and the Middle East[9,10,13,48–50]. To minimize technology-specific batch effects in genotype calls and thus downstream population genetic inference, we focused on previously published ancient samples that had undergone the capture protocol on the same set of SNPs targeted

in our study. We processed these samples through the same pipeline and filters described above, resulting in a reference dataset of 972 ancient samples. Throughout our analysis, we used a subset of $n = 1,088,482$ variants that was created by removing SNPs missing in more than 90% of all ancients individuals (Sardinian and reference dataset) with at least 60% of all captured SNPs covered.

This ancient dataset spans a wide geographic distribution and temporal range (Fig. 2d). For the PCA (Fig. 2a, b), we additionally included a single low-coverage ancient individual (label "Pun") dated to 361–178 BCE from a Punic necropolis on the west Mediterranean island of Ibiza[58]. We merged individuals into groups (Supp. Data 1E, F). For ancient samples, these groups were chosen manually, trying to strike a balance between reducing overlap in the PCA and keeping culturally distinct populations separate. We used geographic location to first broadly group samples into geographic areas (such as Iberia, Central Europe, and Balkans), and then further annotated each of these groups by different time periods.

*Contemporary DNA datasets from Western Eurasia and North Africa*: We downloaded and processed the Human Origins dataset to characterize a subset of Eurasian and North-African human genetic diversity at 594,924 autosomal SNPs[7]. We focused on a subset of 837 individuals from Western Eurasia and North Africa.

*Contemporary DNA dataset from Sardinia*: We merged in a whole-genome sequence Sardinian dataset (1577 individuals[32]) and called genotypes on the Human Origin autosomal SNPs to create a dataset similar to the other modern reference populations. For analyses on province level, we used a subset where at least three grandparents originate from the same geographical location and grouped individuals accordingly (Fig. 2c, $n = 1085$ in total).

**Principal components analysis**. We performed principal components analysis (PCA) on two large-scale datasets of modern genotypes from Western Eurasia and North Africa (837 individuals from the Human Origins dataset) and Sardinia (1577 individuals from the SardiNIA project). For both datasets, we normalized the genotype matrix by mean-centering and scaling the genotypes at each SNP using the inverse of the square-root of heterozygosity[73]. We additionally filtered out rare variants with minor allele frequency ($p_{min} < 0.05$).

To assess population structure in the ancient individuals, we projected them onto the pre-computed principal axes using only the non-missing SNPs via a least-squares approach, and corrected for the shrinkage effect observed in high-dimensional PC score prediction[74] (see Supp. Note 7, Supp. Fig. 23).

We also projected a number of out-sample sub-populations from Sardinia onto our PCs. Reassuringly, these out-of-sample Sardinian individuals project very close to Humans Origins Sardinian individuals (Fig. 2). Moreover, the test-set Sardinia individuals with grand-parental ancestry from Southern Italy cluster with reference individuals with ancestry from Sicily (not shown).

**ADMIXTURE analysis**. We applied ADMIXTURE to an un-normalized genotype matrix of ancient and modern samples[75]. ADMIXTURE is a maximum-likelihood based method for fitting the Pritchard, Stephens and Donnelly model[76] using sequential quadratic programming. We first LD pruned the data matrix based off the modern Western Eurasian and North-African genotypes, using plink1.9 with parameters [–indep-pairwise 200 25 0.4]. We then ran five replicates of ADMIX-TURE for values of $K = 2, …, 11$. We display results for the replicate that reached the highest log-likelihood after the algorithm converged (Supp. Figs. 19–22).

**Estimation of *f*-statistics**. We measured similarity between groups of individuals through computing an outgroup-$f_3$ statistic[77] using the scikit-allel packages's function average_patterson_f3, https://doi.org/10.5281/zenodo.3238280). The out-group-$f_3$ statistic can be interpreted as a measure of the internal branch length of a three-taxa population phylogeny and thus does not depend on genetic drift or systematic error in one of the populations that are being compared[77].

We used the ancestral allelic states as an outgroup, inferred from a multi-species alignment from Ensembl Compara release 59, as annotated in the 1000 Genomes Phase3 sites vcf[78]. We fixed the ancestral allele counts to $n = 10^6$ to avoid finite sample size correction when calculating outgroup $f_3$.

The $f_3$ and $f_4$-statistics that test for admixture were computed with scikit-allel using the functions average_patterson_f3 and average_patterson_d that implement standard estimators of these statistics[77]. We estimated standard errors with a block-jackknife over 1000 markers (blen=1000). For all $f$-statistics calculations, we analyzed only one allele of ancient individuals that were represented as pseudo-haploid genotypes to avoid an artificial appearance of genetic drift—that could for instance mask a negative $f_3$ signal of admixture.

**Estimation of $F_{ST}$-coefficients**. To measure pairwise genetic differentiation between two populations (rather than shared drift from an outgroup as the out-group $f_3$ statistic does), we estimated average pairwise $F_{ST}$ and its standard error via block-jackknife over 1000 markers, using average_patterson_fst from the package scikit-allel. When analyzing ancient individuals that were represented as pseudo-haploid genotypes, we analyzed only one allele. For this analysis, we removed first degree relatives within each population. Another estimator, average_hudson_fst gave highly correlated results ($r^2 = 0.89$), differing mostly for populations with very low sample size ($n \leq 5$) and did not change any qualitative conclusions.

**Estimation of admixture proportions and model testing with qpAdm**. We estimated admixture fractions of a selected target population as well as model consistency for models with up to five source populations as implemented in qpAdm (version 810), which relates a set of "left" populations (the population of interest and candidate ancestral sources) to a set of "right" populations (diverse out-groups)[12]. To assess the robustness of our results to the choice of right populations, we ran one analysis with a previously used set of modern populations as outgroup[12], and another analysis with a set of ancient Europeans that have been previously used to disentangle divergent strains of ancestry present in Europe[50]. In the same qpAdm framework, we use a likelihood-ratio test (LRT) to assess whether a specific reduced-rank model, representing a particular admixture scenario, can be rejected in favor of a maximal rank ("saturated") model for the matrix of $f_4$-values[12]. We report $p$-values under the approximation that the LRT statistic is $\chi^2$ distributed with degrees of freedom determined by the number of "left" and "right" populations used in the $f_4$ calculation and by the rank implied by the number of admixture components. The $p$-values we report are not corrected for multiple testing. Formal correction is difficult as the tests are highly correlated due to shared population data used across them; informally, motivated by a Bonferroni correction of a nominal 0.01 $p$-value with 10 independent tests, we suggest only taking low $p$-values ($< 10^{-3}$) to represent significant evidence to reject a proposed model. The full qpAdm results are discussed in Supp. Note 5.

**Reporting summary**. Further information on research design is available in the Nature Research Reporting Summary linked to this article.

## Data availability

The aligned sequences and processed genotype calls (including read counts) from the data generated in this study are available through the European Nucleotide Archive (ENA, accession number PRJEB35094). Processed read counts and pseudo-haploid genotypes are available via the European Variation Archive (EVA, accession number PRJEB36033) in variant call format (VCF). The contemporary Sardinia data used to support this study have allele frequency summary data deposited to EGA (accession number EGAS00001002212). The disaggregated individual-level sequence data ($n = 1577$) used in this study is a subset of 2105 samples (adult volunteers of the SardiNIA cohort longitudinal study) from Sidore et al. and are available from dbGAP under project identifier phs000313 (v4.p2). The remaining individual-level sequence data originate from a case-control study of autoimmunity from across Sardinia, and per the obtained consent and local IRB, these data are available for collaboration by request from the project leader (Francesco Cucca, Consiglio Nazionale delle Ricerche, Italy).

## Code availability

The code used to process the raw-reads and create the figures in this manuscript can be found at https://github.com/NovembreLab/ancient-sardinia. The code to perform bias correction in predicting out of sample PC scores is publicly available at https://github.com/jhmarcus/pcshrink.

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

## Acknowledgements

We thank Maanasa Rhagavan for in-depth feedback on drafts, Anna Di Rienzo and Goncalo Abecasis for helpful discussions, and Magdalena Zoledziewska for useful comments and early assistance. We would like to thank Antje Wissgott, Cäcilia Freund and other members of the wet laboratory and computational teams at MPI-SHH in Jena. We thank Nadin Rohland, Éadaoin Harney, Shop Mallick, and Alan Cooper for contributing to generating the data for the four samples processed at the Australian Centre for Ancient DNA and in D.R.'s ancient DNA laboratory. We also thank Dan Rice and members of the Novembre lab for helpful discussion and feedback. IRGB-CNR would like to thank the Consortium SA CORONA ARRUBIA DELLA MARMILLA for making available equipment and scientific instruments within the program "Laboratori Dna del Museo Naturalistico del Territorio - Giovanni Pusceddu". This study was supported in part by the Max Planck Society, the University of Sassari, the National Science Foundation via fellowships DGE-1746045 to J.H.M. and DGE-1644869 to T.A.J. and HOMINID grant BCS-1032255 to D.R., the National Institute of General Medical Sciences via training grant T32GM007197 support for J.H.M. and grant RO1GM132383 to J.N., the National Human Genome Research Institute via grant R01HG007089 to J.N., the Intramural Research Program of the National Institute on Aging via contracts N01-AG-1-2109 and HHSN271201100005C to F.C., Fondazione di Sardegna via grants U1301.2015/AI.1157 BE Prat. 2015-1651 to F.C., the Australian Research Council via grant DP130102158 to W.H., the Howard Hughes Medical Institute (D.R.), the University of Pavia INROAd Program (A.O.), the Italian Ministry of Education, University and Research (MIUR) Dipartimenti di Eccellenza Program (2018-2022, A.O.), the Fondazione Cariplo via project 2018-2045 to A.O., the British Academy (R.S.), and the American Society of Prehistoric Research (N.T.).

## Author contributions

We annotate author contributions using the CRediT Taxonomy labels (https://casrai.org/credit/). Where multiple individuals serve in the same role, the degree of contribution is specified as 'lead', 'equal', or 'supporting'. Conceptualization (design of study)—lead: F.C., J.N., J.K., and L.L.; supporting: C.S., C.P., D.S., J.H.M., and G.A. Investigation (collection of skeletal samples)—lead: L.L. and R.S.; supporting: J.B., M.G.G., C.D.S., C.P., V.M., E.P., C.M., A.L.F., D.Ro., M.G., R.P.O., N.T., P.V.D., S.R., P.M., R.B., R.M.S., and P.B. (minor contribution from C.S., J.N.). Investigation (ancient DNA isolation and sequencing)—lead: C.P., A.F., R.R., and M.M.; supporting: C.D.S., W.H., J.K., D.Re*. Data curation (data quality control and initial analysis)—lead: J.H.M., C.P., and H.R.; supporting: C.S., C.C., K.D., H.A., and A.O. Formal Analysis (general population genetics)—lead: J.H.M. and H.R.; supporting: T.A.J. and C.L. Writing (original draft preparation)—lead: J.H.M., H.R., and J.N.; supporting: C.P., R.S., L.L., F.C., and P.V.D. Writing (review and editing)—input from all authors*. Supervision—equal: F.C., J.K., and J.N. Funding acquisition—lead: J.K., F.C., and J.N.; supporting: R.S. *D.R. contributed data for four samples and reviewed the description of the data generation for these samples. As he is also senior author on a separate manuscript that reports data on a non-overlapping set of ancient Sardinians and his group and ours wished to keep the two studies intellectually independent, he did not review the entire paper until after it was accepted.

## Competing interests

The authors declare no competing interests.

## Additional information

Joseph H. Marcus [1,34], Cosimo Posth [2,3,34], Harald Ringbauer [1,34], Luca Lai[4,5], Robin Skeates[6], Carlo Sidore [7,8], Jessica Beckett[9], Anja Furtwängler[3], Anna Olivieri [10], Charleston W.K. Chiang [11,12], Hussein Al-Asadi[13,14], Kushal Dey[13,15], Tyler A. Joseph[16], Chi-Chun Liu[1], Clio Der Sarkissian[17], Rita Radzevičiūtė[2], Megan Michel[2,18], Maria Giuseppina Gradoli[19], Patrizia Marongiu[8], Salvatore Rubino[8], Vittorio Mazzarello[8], Daniela Rovina[20], Alessandra La Fragola[21], Rita Maria Serra[8,22], Pasquale Bandiera [8,22], Raffaella Bianucci[23,24], Elisa Pompianu[25], Clizia Murgia[26], Michele Guirguis [25], Rosana Pla Orquin [25], Noreen Tuross [18], Peter van Dommelen [27], Wolfgang Haak [2], David Reich [28,29,30,31], David Schlessinger[32], Francesco Cucca[7,8 ✉], Johannes Krause [2,3,31 ✉] & John Novembre [1,33 ✉]

[1]Department of Human Genetics, University of Chicago, Chicago, IL, USA. [2]Max Planck Institute for the Science of Human History, Jena, Germany. [3]Institute for Archaeological Sciences, University of Tübingen, Tübingen, Germany. [4]Department of Anthropology, University of South Florida, Tampa, FL, USA. [5]Department of Anthropology, University of North Carolina at Charlotte, Charlotte, NC, USA. [6]Department of Archaeology, Durham University, Durham, UK. [7]Istituto di Ricerca Genetica e Biomedica - CNR, Cagliari, Italy. [8]Dipartimento di Scienze Biomediche, Università di Sassari, Sassari, Italy. [9]Private contractor, Cagliari, Sardinia, Italy. [10]Dipartimento di Biologia e Biotecnologie "L. Spallanzani", Università di Pavia, Pavia, Italy. [11]Center for Genetic Epidemiology, Department of Preventive Medicine, Keck School of Medicine, University of Southern California, Los Angeles, CA, USA. [12]Quantitative and Computational Biology Section, Department of Biological Sciences, University of Southern California, Los

Angeles, CA, USA. [13]Department of Statistics, University of Chicago, Chicago, IL, USA. [14]Committee on Evolutionary Biology, University of Chicago, Chicago, IL, USA. [15]Department of Epidemiology, Harvard School of Public Health, Boston, MA 02115, USA. [16]Department of Computer Science, Columbia University, New York, NY, USA. [17]Laboratoire d'Anthropologie Moléculaire et d'Imagerie de Synthèse, CNRS UMR 5288, Université de Toulouse 3, Toulouse, France. [18]Department of Human Evolutionary Biology, Harvard University, Cambridge, MA 02138, USA. [19]School of Archaeology and Ancient History, University of Leicester, Leicester, UK. [20]Soprintendenza Archeologia, belle arti e paesaggio delle province di Sassari e Nuoro, Sassari, Italy. [21]Departamento de Geografía, Historia y Humanidades Escuela Internacional de Doctorado de la Universidad de Almería, Almería, Spain. [22]Center for Anthropological, Paleopathological and Historical Studies of the Sardinian and Mediterranean Populations, University of Sassari, Sassari, Italy. [23]Department of Sciences and Technological Innovation, University of Eastern Piedmont, 15121 Alessandria, Italy. [24]Legal Medicine Section, Department of Public Health and Paediatric Sciences, University of Turin, 10126 Turin, Italy. [25]Department of History, Human Sciences and Education, University of Sassari, 07100 Sassari, Italy. [26]Universitat Autònoma de Barcelona, Departament de Biologia Animal, Biologia Vegetal i Ecologia, 08193 Barcelona, Spain. [27]Joukowsky Institute for Archaeology and the Ancient World, Brown University, Providence, RI 02912, USA. [28]Department of Genetics, Harvard Medical School, Boston, MA 02115, USA. [29]Broad Institute of Harvard and MIT, Cambridge, MA, USA. [30]Howard Hughes Medical Institute, Harvard Medical School, Boston, MA, USA. [31]Max Planck-Harvard Research Center for the Archaeoscience of the Ancient Mediterranean, Munich, Germany. [32]Laboratory of Genetics, NIA, NIH, Baltimore, MD, USA. [33]Department of Ecology and Evolution, University of Chicago, Chicago, IL, USA. [34]These authors contributed equally: Joseph H. Marcus, Cosimo Posth, Harald Ringbauer. ✉email: fcucca@uniss.it; krause@shh.mpg.de; jnovembre@uchicago.edu

