## [Peer Review File · Nature Communications]

Reviewers' Comments:

Reviewer #1:

Remarks to the Author:

In the manuscript entitled "Population history from the Neolithic to present on the Mediterranean island of Sardinia: An ancient DNA perspective", Marcus et al. describes the genomic analysis of ancient samples from Sardinia, including three time periods: Neolithic/Early Copper Age, Early Middle Bronze Age and the Nuragic period. By comparing their ancient samples to available aDNA data, they aim to test the hypothesis that modern Sardinian DNA descends directly from Neolithic people and that Sardinia was not affected by later migration associated to the Steppe expansion. Marcus et al. find that all ancient samples from Sardinia present high affinity with early Neolithic people from Europe and lack the Steppe component. On the other hand, modern Sardinians show the influence of later migrations from eastern and northern Mediterranean areas.

I think that the manuscript is interesting and within the scope of Nature Communications. This study provides new ancient DNA data from a very interesting area and it is a good example of how aDNA can be used to disentangle complex migration patterns. In my opinion, lab procedures meet the standard for ancient DNA analysis, the manuscript is well written and the conclusions are supported by extensive data analyses.

I find particularly enjoyable the detailed analysis of Y-chromosome results, which are sometimes discussed very briefly in major aDNA publications. Particularly, the results on R-V88 are very interesting and provide a fresh look at the lineage's origin based on aDNA data.

I just have a couple of minor comments/suggestions:

- Main text, Results section: If Mediterranean expansions affected Sardinia, it is possible they received some North African ancestry. Why using just North African Jews for qpAdm? Apart from the analysis using aDNA from Guanches and Moroccan Neolithic people, did you try using modern North African populations?
- Supplementary Information 3, page 4: Could you explain the implications of aRchaic plots further? Because this method is still not widely applied to test for DNA damage, its interpretation could not be evident for readers. As expected no_UDG and UDG ancient samples are different from the modern ones, but what is causing the two clusters in the modern DNA?
- Supplementary Information 4, page 6: remove extra parenthesis in "(...) Pyrenees (modern Aragon, Spain)), (...)".
- Supplementary Information 6: I have problems visualizing Figure S11, the figure seems incomplete.

Rosa Fregel

Reviewer #2:

Remarks to the Author:

Marcus, Posth, Ringbauer et al. present the first genomic study of prehistoric individuals from the Mediterranean island of Sardinia. The relatively isolated population of modern Sardinians has been a focus of genetic studies for a long time. For ancient DNA researchers, this island is of particular interest as ancient European Neolithic farmers usually show the highest genetic similarity to modern Sardinians. This manuscript adds another dimension to this pattern by presenting actual genomic data from Neolithic Sardinian farmers. Furthermore, they provide additional data from later periods and compare the ancient individuals to a large reference data set of modern

Sardinian genomes.

I do not have any major concerns about this manuscript. It reads nicely and the main conclusions are based on established methodology. I have some minor comments and recommendations that I am outlining below. Since I am not an expert on Sardinian archaeology, I am not going to comment on that part of the study.

One point I need to comment on is chronology. In the abstract, the authors write that their study “sheds new light on the origin of the Neolithic settlement on Sardinia” but in the rest of the text, they do not directly show anything about the “origin”. First, they do not have any early Neolithic individuals from Sardinia which would represent the direct descendants of some continental early Neolithic group. Second, there is not enough reference data to represent all potential source regions. I would suggest to revise the abstract. The fact that only Middle Neolithic individuals from Sardinia are studied restricts the ability to draw conclusions on the first Neolithic migrants. The Sardinian individuals are mainly compared to mainland early Neolithic groups which are ~1000 years older and which also show a lower proportion of WHG ancestry. The main change in WHG ancestry happens between EN and MN on the mainland so it is not clear whether the MN Sardinians have this higher WHG proportion due to local admixture or because they come from a different source than the EN groups sequenced so far. This could also explain the contradiction between f statistics identifying Iberians as potential source and qpAdm identifying French as best fitting source (the latter could be just due to similar admixture proportions). The current data does not even allow to conclude whether Sardinia was part of the original Mediterranean expansion or some later migration. The authors should be more specific throughout the manuscript that their conclusions are affecting the middle Neolithic on Sardinia and subsequent periods but not the early Neolithic. Furthermore, I would suggest to repeat the analysis after splitting the NECA group into two. The sample size is small but this group currently spans more than 1000 years with a large gap which might mask differences over time.

One connection that the authors do not explore in depth is African ancestry on Sardinia. They only use modern Tunisians, pre-contact Canary Islanders and Neolithic Moroccans as potential sources for African ancestry. One could also test Berbers like the Mozabite or other ancient north Africans as potential sources. The main reason why I am pointing this out is that their previous article on modern Sardinian genomes even suggested a connection to sub-Saharan Africa and the parallel Article on ancient Mediterranean islanders also finds some African ancestry.

When I first saw the manuscript and its supplementary material, I wanted to applaud the authors for going beyond the standard PCA/Admixture/ f statistics scheme of ancient DNA studies these days. For example, the authors use both f statistics and F_{st} , they use a novel method on clustering sequenced reads, they run a new implementation of ADMIXTURE that takes sampling times into account, and they compare different PCA projection methods. However, these results mostly end up in the supplementary material. I know this might not be the focus of this manuscript but one could easily investigate this further which would make this study more interesting from a methodological standpoint. For example: Why were F_{st} and outgroup f_3 calculated and how do the results differ (and why)? How do the Dystruct results relate to the qpAdm results?

The authors are aware of a parallel study that investigates a non-overlapping set of individuals from Sardinia and other Mediterranean islands which has at least one overlapping author. I understand why they would keep these studies separate and I encourage that to value the work invested by numerous trainees. However, since both articles are on biorxiv, the authors should cite the other work and compare their results.

Cosmetic comments:

- Maybe quantify the proportion of continuity since the Neolithic or Neolithic Anatolian ancestry in modern Sardinians and use those values in the abstract. I know this is not straightforward as

many of the later contributions probably had high proportions of EEF ancestry as well.

- L68: Please cite the proper articles which showed these geographic regions for the first time. Some are cited elsewhere in the manuscript, others are not (e.g. Gamba et al 2014, Olalde et al 2015).
- I really like the idea behind Fig 1A but I would suggest to re-plot figure 1B. The symbols could have thinner outlines (e.g. like in 1A) and different symbol shapes could be used as well.
- In the notation for f_4 statistics, the authors use a "-" which can be misinterpreted as a subtraction of allele frequencies, I suggest to replace them with a comma.
- Fig 2B and 4A: please show single individuals as dots, the figures become quite busy with all of the text.
- L258-263: Please be more specific. You are talking about affinity with "some" populations, please name them or give examples. From the text it is also not clear what kind of admixture the f_3 statistics suggest. Fig 4 only shows Karitiana.
- The term "Northern Mediterranean" is not very well defined in my opinion, especially if it includes Malta and Sicily (which are geographically more southern or central). I would suggest to name all populations included under this term at the first mentioning of it.
- The text refers to figure 4 in many occasions. Please always add what sub-figure you are referring to. What are the numbers in the circles of B and C (since the f statistic seems to be shown in the color scale)? Please explain the motivation for using Karitiana in the text as well. Do the conclusions change when using Yamnaya?
- Please list the versions of all softwares used.
- The methods do only describe what tool was used for admixture f_3 but not how outgroup f_3 was calculated.
- f_4 statistics were calculated with a module for D statistics. Does this module need specific settings to run f_4 ?
- Somehow the formula in line 598 seems to have failed.
- Why is the supplementary text split in two files?

Supplement:

- Section 3: What happens when aRchaic is run with 2 or 3 clusters? The original article presenting this method also tested whether it could be used to estimate contamination. Would the results be consistent with other contamination estimates in this study?
 - Figure 3: The vertical jitter is a bit problematic since it places many symbols into neighboring haplogroups.
 - Section 4: Maybe one could show a small phylogeny of haplogroups I and R. The usage of the markers only can be hard to follow for someone not very familiar with Y chromosomes as it might mask that some of these are subgroups of others. It would also be nice to show more information on the mutations underlying the classifications than just listing single markers in a supplementary table.
- Figure 15: I think both blue and purple represent EEF ancestry while red corresponds to WHGs (in contrast to the figure legend).

Reviewer #3:

Remarks to the Author:

Marcus et al presents novel genome-wide data from a temporal transect of 43 ancient Sardinians to investigate the population history of the island. Since the publication of the genome of Otzi, Sardinians have held a special interest in ancient DNA studies of prehistoric Europeans, due to their high genetic affinity to individuals from a "early European farmer" context. This is the first study to directly investigate the genetic ancestry of Sardinian individuals during and after the Neolithic, and as such a major contribution to our understanding of European pre-history.

The major findings of the study are

- Strong genetic affinity with western Mediterranean Neolithic populations
- Continuity from the Neolithic to the Nuragic period, with no evidence of arrival of Steppe ancestry seen across mainland Europe
- Moderate gene flow into the island after 1000 BCE, inferred to originate from both northern and eastern Mediterranean sources

I found this to be an exceptionally clear manuscript, well written and with highly informative figures. The methodologies applied are following the established protocol for ancient population genetics studies, from genetic clustering using PCA and ADMIXTURE to more formal tests for admixture using the extended f-statistic framework. The major results and conclusions presented are well supported by these analyses and adequately documented, and as such I only have some more minor follow-up questions/suggestions on some of the specific results.

Genetic continuity

The authors make a very convincing claim on genetic continuity on Sardinia over the 3,000 year transect from the Neolithic to the Bronze Age in their dataset, with no apparent temporal signal of admixture in their data. However, there is quite strong geographic stratification in the distribution of Y chromosome haplogroups between the regions, with I2-M223 dominating in the northeast and R1b-V88 more frequent in the south-central region between Oristano and Cagliari. I am curious as to whether there might be evidence for geographic structure also when using the autosomal data. One of the strengths of their data is the availability of a large number of present-day Sardinians to investigate local structure more in detail. There is an analysis to that end projecting the ancient individuals on the first two PCs in figure 4A, which suggests limited structure. However, those are largely driven by the highly drifted Ogliastro individuals, is there any further differentiation among the other regions observed in the higher PCs? Given the large number of ancient samples available another possibility would be to directly cluster them without projecting onto modern individuals, using an appropriate pairwise distance metric.

Post-Nuragic admixture

The comparison with modern Sardinians suggests moderate levels of post-Nuragic admixture, with >40% non-Nuragic contribution even in the more isolated Ogliastro populations. This turnover is also reflected on the Y-chromosome, with the major ancient haplogroups I2-M223 and R1b-V88 much rarer among contemporary Sardinians. Overall, I enjoyed the discussion of these results in the context of affinities with early farmers, and their tying together with similar results observed in Basques, which provides a nice resolution of some long-standing questions in European pre-history. Nevertheless, the amount of turnover across the entire island is somewhat surprising, so I would be interested to see whether any more insights could be made into its timing. Given that Nuragic individuals are good proxy population for at least the Sardinian source population, a possibility would be to test whether e.g. ALDER would yield an admixture LD curve for dating it using modern Sardinians as targets.

Minor edits

Figure 1 – Including the names of sampling locations for the ancient individuals in the inset would be helpful.

Supplementary information 4 – The text mentions the “Sardinian” haplogroup I2-M26, but in figure S3 this is labelled as I2-L160.

Fig S15 - The colours in the legend are mislabelled/switched for WHG (purple) and Middle East (red)

RESPONSE TO REVIEWS NCOMMS-19-08622

General note for reviewers: Thanks for your comments. For many of the major responses, we have highlighted the changed text in the pdf and added a footnote with a numeric reference to the relevant reviewer comment. To avoid visual clutter, we did not mark minor wording edits throughout nor obvious changes that occurred to incorporate the new data.

Reviewer #1 (Remarks to the Author):

[...] I think that the manuscript is interesting and within the scope of Nature Communications. This study provides new ancient DNA data from a very interesting area and it is a good example of how aDNA can be used to disentangle complex migration patterns. In my opinion, lab procedures meet the standard for ancient DNA analysis, the manuscript is well written and the conclusions are supported by extensive data analyses.

I find particularly enjoyable the detailed analysis of Y-chromosome results, which are sometimes discussed very briefly in major aDNA publications. Particularly, the results on R-V88 are very interesting and provide a fresh look at the lineage's origin based on aDNA data.

Thanks for your positive comments.

I just have a couple of minor comments/suggestions:

[1.1] Main text, Results section: If Mediterranean expansions affected Sardinia, it is possible they received some North African ancestry. [a] Why using just North African Jews for qpAdm? [b] Apart from the analysis using aDNA from Guanches and Moroccan Neolithic people, did you try using modern North African populations?

R1.1a/b: The point is very well taken that the topic deserves more attention, and is echoed by Reviewer 2's commentary (R2.2). In the new analysis:

- 1) We included several more present-day North African populations from the HO dataset (Mozabite, Saharawi, Egyptian) In particular, we included these individuals in calculating the main PCA (Fig. 2) and in unsupervised clustering with Admixture (Fig. 5), which improves the sensitivity to North African ancestry within these analysis. In the original qpAdm analyses we had only used North African Jewish populations, a few Levant populations, and Tunisians from N. Africa.**
- 2) We included more ancient Moroccan samples (Morocco-EN and Morocco-Iberomaurusian), which, to our knowledge, are the only existing genome-wide ancient samples from this region. We included these in the Admixture analysis, and now observe one cluster that is maximized in ancient North Africans for k= 6 (Fig. 5).**

- 3) **Using the Morocco-EN samples, we run a 5-way admixture model (Supp. Fig. 11) and 4-way admixture model with Morocco-EN as outgroup (Supp. Fig. 11). We use Morocco-EN for these analyses as they are less admixed in comparison to Morocco-LN, which seem to carry about 50% Early European Farmer-related ancestry and thus make some results more complicated to interpret.**

Corroborating patterns found in previous papers (and the related Fernandez et al pre-print), we find evidence for only a small amount of North African admixture in present-day Sardinian, and negligible amounts in the ancient sample. An exception is the Punic site Villamar, which seems to carry more significant North African ancestry. These new results are summarized in the results and a paragraph in the discussion.

- [1.2] Supplementary Information 3, page 4: [a] Could you explain the implications of aRchaic plots further? Because this method is still not widely applied to test for DNA damage, its interpretation could not be evident for readers. [b] As expected no_UDG and UDG ancient samples are different from the modern ones, but what is causing the two clusters in the modern DNA?

Yes - we apologize that the initial descriptions of these results were too brief. [1.2a:] We have now added text to the description of the aRchaic results in the supplement to help explain the implications of the results more clearly. [1.2b:] We are unsure what is causing the two clusters in the modern DNA, but suspect that it has something to do with library preparation or initial DNA quality – in the original aRchaic paper clusters were observed that were due to library prep protocols. We now discuss this in the extended description.

- [1.3] Supplementary Information 4, page 6: remove extra parenthesis in “(...) Pyrenees (modern Aragon, Spain)), (...)”.

We have fixed this typo.

- [1.4] Supplementary Information 6: I have problems visualizing Figure S11, the figure seems incomplete.

We apologize for the confusion. The blank areas indicated models for which qpAdm did not produce viable results (admixture fractions outside [0,1]). In the new figures, we

have plotted a gray bar for these infeasible models, and describe this approach in the Figure caption.

Rosa Fregel

Reviewer #2 (Remarks to the Author):

Marcus, Posth, Ringbauer et al. present the first genomic study of prehistoric individuals from the Mediterranean island of Sardinia. The relatively isolated population of modern Sardinians has been a focus of genetic studies for a long time. For ancient DNA researchers, this island is of particular interest as ancient European Neolithic farmers usually show the highest genetic similarity to modern Sardinians. This manuscript adds another dimension to this pattern by presenting actual genomic data from Neolithic Sardinian farmers. Furthermore, they provide additional data from later periods and compare the ancient individuals to a large reference data set of modern Sardinian genomes.

I do not have any major concerns about this manuscript. It reads nicely and the main conclusions are based on established methodology. I have some minor comments and recommendations that I am outlining below. Since I am not an expert on Sardinian archaeology, I am not going to comment on that part of the study.

[R2.1] One point I need to comment on is chronology. In the abstract, the authors write that their study “sheds new light on the origin of the Neolithic settlement on Sardinia” but in the rest of the text, they do not directly show anything about the “origin”. [R2.1a] First, they do not have any early Neolithic individuals from Sardinia which would represent the direct descendants of some continental early Neolithic group.

We appreciate the reviewer’s point. Our earliest samples are indeed from the Middle Neolithic rather than the early Neolithic. We are now careful to designate our earliest samples as from the “Middle Neolithic” to emphasize that we do not have samples from the early Neolithic as the reviewer noted. We have struck the phrase: “on the origin of Neolithic settlement on Sardinia” from the abstract. Also to help, but not resolving the issue, we were able to obtain additional samples from the Neolithic, improving our sample size from 2 to 6 and pushing back our oldest sample slightly (6389-6210 cal BP).

[R2.1b] Second, there is not enough reference data to represent all potential source regions. I would suggest to revise the abstract. The fact that only Middle Neolithic individuals from Sardinia are studied restricts the ability to draw conclusions on the first Neolithic migrants. The Sardinian individuals are mainly compared to mainland early Neolithic groups which are ~1000 years older and which also show a lower proportion of WHG ancestry. The main change in WHG ancestry happens between EN and MN on the mainland so it is not clear whether the MN Sardinians have this higher WHG proportion due to local admixture or because they come from a different source than the EN groups sequenced so far.

We had acknowledged the caveat about potential unsampled source regions briefly in the original manuscript, however we now emphasize the uncertainty more emphatically, and work in more strongly the reviewer's good point about the difference in WHG ancestry in Sardinia versus the mainland. The key changes are in the revised first paragraphs of the Discussion section.

[R2.1c] This could also explain the contradiction between f statistics identifying Iberians as potential source and qpAdm identifying French as best fitting source (the latter could be just due to similar admixture proportions). The current data does not even allow to conclude whether Sardinia was part of the original Mediterranean expansion or some later migration. The authors should be more specific throughout the manuscript that their conclusions are affecting the middle Neolithic on Sardinia and subsequent periods but not the early Neolithic.

We are now careful to designate our earliest samples as from the "Middle Neolithic" to emphasize that we do not have samples from the early Neolithic as the reviewer noted. Regarding the French as a best fitting source, we now specify that the French may fit because of similar admixture proportions as suggested by the reviewer, and we have softened our discussion of the results to further emphasize the uncertainty that arises by not having samples from mainland Italy.

[R2.1d] Furthermore, I would suggest to repeat the analysis after splitting the NECA group into two. The sample size is small but this group currently spans more than 1000 years with a large gap which might mask differences over time.

In the revised paper we now divide the sample into a Middle to Late Neolithic group (Sar-MN, n=8), an Early Copper Age group (Sar-ECA, n=3), and Early Middle Bronze Age

groups (Sar-EMBA, n= 27). All three periods have remarkably similar levels of admixture proportions (Table 1). This suggests the previous grouping was not obscuring meaningful differences. It was very helpful to verify this, and the finer-scale groupings hopefully improve the paper.

[R2.2] One connection that the authors do not explore in depth is African ancestry on Sardinia. They only use modern Tunisians, pre-contact Canary Islanders and Neolithic Moroccans as potential sources for African ancestry. One could also test Berbers like the Mozabite or other ancient north Africans as potential sources. The main reason why I am pointing this out is that their previous article on modern Sardinian genomes even suggested a connection to sub-Saharan Africa and the parallel Article on ancient Mediterranean islanders also finds some African ancestry.

See Response to R1.1 above.

[R2.3] When I first saw the manuscript and its supplementary material, I wanted to applaud the authors for going beyond the standard PCA/Admixture/fstatistics scheme of ancient DNA studies these days. For example, the authors use both f statistics and F_{st} , they use a novel method on clustering sequenced reads, they run a new implementation of ADMIXTURE that takes sampling times into account, and they compare different PCA projection methods.

[R2.3a] However, these results mostly end up in the supplementary material. I know this might not be the focus of this manuscript but one could easily investigate this further which would make this study more interesting from a methodological standpoint. [R2.3b] For example: Why were F_{st} and outgroup f_3 calculated and how do the results differ (and why)? [R2.3c] How do the Dystruct results relate to the qpAdm results?

We appreciate you were able to see the efforts made to test different methodologies and regret we have found it difficult to incorporate more discussion of methodologies in the manuscript. That said, now: [R2.3a]: We have added a comment about PCA projection into the main text, especially as we found using shrinkage correction in PCA projection to be quite important and want to direct readers to the Supp Info section regarding it. [R2.3b]: We computed both f_{st} and outgroup f_3 since the former is sensitive to drift within one of the two populations being compared, whereas outgroup f_3 is much less so. This qualitative difference allows one to identify populations which could have experienced localized genetic drift (such as observed for instance for Ogliastra, as

evidenced by its high F_{st} with other populations). We added a brief explanation to Materials & Methods. [R2.3c]: For brevity, we chose to leave the existing discussion comparing the ADMIXTURE and Dystruct results in the Supplement.

[R2.4] The authors are aware of a parallel study that investigates a non-overlapping set of individuals from Sardinia and other Mediterranean islands which has at least one overlapping author. I understand why they would keep these studies separate and I encourage that to value the work invested by numerous trainees. However, since both articles are on biorxiv, the authors should cite the other work and compare their results.

In the revised manuscript, we now cite the other paper and carry out comparative analyses to theirs. Generally our work agrees with theirs (as highlighted in the new discussion). For the individuals from the Nuragic period, they find support for a two-way model with WHG and EEF ancestry, and infer WHG ancestry close to 20 percent, with no evidence for Steppe or other distinct ancestries. They report the arrival of both Iranian and Steppe ancestry on the island by Late antiquity, and we observe a similar signal, extending as early as to individuals from Phoenician/Punic sites. Their inferred Y haplogroups are a close match to the haplogroups we detected, with R1b and G2 subclades in Sardinians from the Nuragic period.

Notably, the Fernandez et al preprint presents results for a model in which present-day Sardinians are modeled as derived from 13% Nuragic ancestry and 86% from ancestry represented by a Late Antiquity Sardinian, who themselves can be modeled as having essentially no Nuragic ancestry. We are cautious about interpreting those results as indicating modern Sardinians necessarily have low Nuragic ancestry, because in our own analyses, we found a wide range of estimates emerge for the proportion of Nuragic ancestry in post-Nuragic ancient individuals depending on what outgroups and ingroups were used (even as we replicate that some individuals can be fit without Nuragic ancestry). We suspect some of this uncertainty emerges in part because the relatively low differentiation among Nuragic period and mainland samples makes it challenging to carry out precise inferences, especially using individual-level analysis.

Interestingly, we find that for the Corona Moltana (COR) individuals and the ORC002 individual, models without Nuragic ancestry fail, and models of one-way continuity from these individuals to the present cannot be rejected. Also the PCA placements of the COR

samples are the only ones overlapping with core modern-day Sardinians and not on top of Nuragic and pre-Nuragic people. In other words, the signatures of continuity from the Nuragic period are heterogeneous across post-Nuragic sites. We show that, as observed in modern Sardinia samples, post-Nuragic Sardinia was spatially and temporally heterogeneous. Until a larger sampling is available, the exact fine-scale processes of contact with the mainland will remain uncertain. However, both studies make clear models of Sardinian isolation from the Neolithic through the present need adjustment to account for the enhanced diversity in ancestry that has characterized the island since the Iron Age. Our new discussion tries to bring this complexity to the reader.

Cosmetic comments:

- [R2.5] Maybe quantify the proportion of continuity since the Neolithic or Neolithic Anatolian ancestry in modern Sardinians and use those values in the abstract. I know this is not straightforward as many of the later contributions probably had high proportions of EEF ancestry as well.

We estimate EEF ancestry in modern Sardinians in Table 1, but have chosen not to include the proportion in the abstract. In general, we try to avoid putting numbers into the abstract that can be misinterpreted without the appropriate caveats that can be explained at more length in the discussion.

- [R2.6] L68: Please cite the proper articles which showed these geographic regions for the first time. Some are cited elsewhere in the manuscript, others are not (e.g. Gamba et al 2014, Olalde et al 2015).

This was an oversight. We have now added the appropriate citations.

- [R2.7] I really like the idea behind Fig 1A but I would suggest to re-plot figure 1B. The symbols could have thinner outlines (e.g. like in 1A) and different symbol shapes could be used as well.

We thank the reviewer for pointing out the discrepancy in symbol outlines between figures 1A and 1B. We have updated the figures so that the ratio of symbol to outline are precisely consistent.

[R.2.8] In the notation for f4 statistics, the authors use a “-” which can be misinterpreted as a subtraction of allele frequencies, I suggest to replace them with a comma.

The use of a minus is intentional. This notation helps to make it more clear what f4 statistic is being calculated (as the f4 statistic is a product of differences in allele frequencies its natural to show the subtraction sign). However, the typesetting was suboptimal, and we have tried to improve it now.

[R.2.9] - Fig 2B and 4A: please show single individuals as dots, the figures become quite busy with all of the text.

We appreciate the issue the reviewer raises and struggle with these decisions; however, while the visual clutter prevents reading the labels in some places, in general we find text labels contain more information and eases the cognitive burden over using dozens of point types or simple dots that contain no sub-population information without reference to large legends.

[R.2.10] - L258-263: Please be more specific. You are talking about affinity with “some” populations, please name them or give examples. From the text it is also not clear what kind of admixture the f3 statistics suggest. Fig 4 only shows Karitiana.

We modified this section to tighten the writing and avoid the confusion. The qpAdm analysis is based on f4 statistics, and we believe it provides a more interpretable overview of the results. We therefore put the focus of the main text on these. We still report all f4 and f3 as a Supplementary Table. For Figure 4A, we substituted a zoomed of the Figure 2 PCA, which conveys the same conclusions while being more interpretable.

[R2.11] - The term “Northern Mediterranean” is not very well defined in my opinion, especially if it includes Malta and Sicily (which are geographically more southern or central). I would suggest to name all populations included under this term at the first mentioning of it.

We now list the populations tested in the text when we first use the term “northern” and “eastern Mediterranean”.

[R2.12] - [a] The text refers to figure 4 in many occasions. Please always add what sub-figure you are referring to. [b] What are the numbers in the circles of B and C (since the f statistic seems to be shown in the color scale)? [c] Please explain the motivation for using Karitiana in the text as well. Do the conclusions change when using Yamnaya?

We'll respond out of order as it's helpful here. First [c]: The results do not quantitatively change using the Yamnaya, though the magnitude of the f4 statistics becomes weaker. We continue to report these f4 statistics for Cagliari/Olbia-Tempio as well as Ogliastra for a big set of outgroup populations, including Steppe-EMBA which is a grouping of Steppe/Yamanya ancestry individuals in Supp. Mat. 2. That said, we realized the original panels B and C were confusing because of this specific choice and therefore decided to remove these figures. Instead we now show a zoom in on the Figure 2 PCA highlighting within Sardinia diversity, which addresses the same question of differential ancestry patterns in present-day Sardinia based on a similar signal, and should be more readily interpretable.

[a]: We apologize and have revised the text to add the appropriate sub-figure whenever citing figures with multiple sub-panels ..

[b]: The numbers depicted the f-statistic values and the standard deviations; that is the numbers and colors were redundant. That said, the issue does not matter as that figure panel has been removed.

[R2.13]- Please list the versions of all softwares used.

This information is provided in the Nature Communications "Reporting Form" on reproducibility that we understand will be included alongside the published paper.

[R2.14]- The methods do only describe what tool was used for admixture f3 but not how outgroup f3 was calculated.

We have now added the function we used for these calculations in the description in Materials and Methods (*average_patterson_f3*). We now also cite the paper that introduced this estimator (Patterson 2012).

[R2.15]- f4 statistics were calculated with a module for D statistics. Does this module need specific settings to run f4?

This module (*average_patterson_d*) returns the estimator for f4 statistics $f_4(P_1-P_2; P_3-P_4) = E(p_1-p_2, p_3-p_4)$ as part of its output. The only required parameter in addition to the allele counts is the block length for block bootstrapping, and we report that.

[R2.16]- Somehow the formula in line 598 seems to have failed.

We apologize but the original submission does not seem to have a formula on line 598 or in that section generally. To address this comment though, we have proofread the typesetting of all formulas.

[R2.17]- Why is the supplementary text split in two files?

This was a minor issue of book-keeping - the archaeology site descriptions were typeset in Word and the remainder in LaTeX and hence we have them in two files. We have now merged them into a single document.

[R2.18] Supplement:

- Section 3: [R2.18a] What happens when aRchaic is run with 2 or 3 clusters? [R2.18b] The original article presenting this method also tested whether it could be used to estimate contamination. Would the results be consistent with other contamination estimates in this study?

[R 2.18a] We ran aRchaic, presetting the number of clusters to K=2 and K=3 and updated the supplementary section to reflect these additions (Supplemental Info Section 2, Supp Fig. 4, Supp Fig. 5). When we increase K we see additional substructure revealed amongst the sample. For K=2 the two mismatch profiles clearly delineate ancient and modern individuals. When we set K=3 we observe structure amongst ancient individuals, particularly ancient individuals whose sequencing libraries were treated UDG treatment (UDG) or not treated (no UDG) primarily have membership in different clusters. As expected the mismatch profile that are represented by individuals with the UDG treatment has a reduced signal of the ancient DNA damage signature.

[R 2.18b] We only ran aRchaic on samples that passed our mtDNA and X-based contamination filters -- so we a priori expect this set of samples to have low

contamination. In agreement, we find none of then bam files show a significant mixture proportion from the modern damage profile clusters.

[R2.19] - Figure 3: The vertical jitter is a bit problematic since it places may symbols into neighboring haplogroups.

We reduced the vertical jitter in this figure. Now every individual effectively overlaps with its haplogroup. For the haplogroups that are rare in the present, there is still some visual ambiguity left, but we hope that the haplogroup colors provide the reader sufficient information to resolve this uncertainty.

[R2.20] - Section 4: Maybe one could show a small phylogeny of haplogroups I and R. The usage of the markers only can be hard to follow for someone not very familiar with Y chromosomes as it might mask that some of these are subgroups of others. It would also be nice to show more information on the mutations underlying the classifications than just listing single markers in a supplementary table.

We now visualize the topology of key R and I haplogroups, and we report the number of equivalent markers that intersect the 1240k target (Supp. Fig. 7). For the full tree, as well as complete descriptions of all defining mutations used within this work we refer to the full tables of ISOGG 2018 (which together with the Y readcount data we will report should provide full reproducibility).

[R2.21] Figure 15: I think both blue and purple represent EEF ancestry while red corresponds to WHGs (in contrast to the figure legend).

We have modified the Figure caption to describe the correct colors.

Reviewer #3 (Remarks to the Author):

Marcus et al presents novel genome-wide data from a temporal transect of 43 ancient Sardinians to investigate the population history of the island. Since the publication of the genome of Otzi, Sardinians have held a special interest in ancient DNA studies of prehistoric

Europeans, due to their high genetic affinity to individuals from a “early European farmer” context. This is the first study to directly investigate the genetic ancestry of Sardinian individuals during and after the Neolithic, and as such a major contribution to our understanding of European prehistory.

The major findings of the study are

- Strong genetic affinity with western Mediterranean Neolithic populations
- Continuity from the Neolithic to the Nuragic period, with no evidence of arrival of Steppe ancestry seen across mainland Europe
- Moderate gene flow into the island after 1000 BCE, inferred to originate from both northern and eastern Mediterranean sources

I found this to be an exceptionally clear manuscript, well written and with highly informative figures. The methodologies applied are following the established protocol for ancient population genetics studies, from genetic clustering using PCA and ADMIXTURE to more formal tests for admixture using the extended f-statistic framework. The major results and conclusions presented are well supported by these analyses and adequately documented, and as such I only have some more minor follow-up questions/suggestions on some of the specific results.

Genetic continuity

[R3.1] The authors make a very convincing claim on genetic continuity on Sardinia over the 3,000 year transect from the Neolithic to the Bronze Age in their dataset, with no apparent temporal signal of admixture in their data. However, there is quite strong geographic stratification in the distribution of Y chromosome haplogroups between the regions, with I2-M223 dominating in the northeast and R1b-V88 more frequent in the south-central region between Oristano and Cagliari. I am curious as to whether there might be evidence for geographic structure also when using the autosomal data. One of the strengths of their data is the availability of a large number of present-day Sardinians to investigate local structure more in detail. There is an analysis to that end projecting the ancient individuals on the first two PCs in figure 4A, which suggests limited structure. [R3.1a] However, those are largely driven by the highly drifted Ogliastra individuals, is there any further differentiation among the other regions observed in the higher PCs? [R3.b] Given the large number of ancient samples available another possibility would be to directly cluster them without projecting onto modern individuals, using an appropriate pairwise distance metric.

R3.1a : We agree Figure 4A was dominated by the drift of the Ogliastro individuals, and in this revision we double checked that this not due to the large number of individuals sampled from there (by down-sampling the number of individuals from Ogliastro as well as Cagliari to obtain more even sample sizes across the whole).

To more formally explore the issue of whether there is subtle structure that appears in lower PCs of the within-Sardinia analysis, we have added a new Supplementary Information section of analyses (Section 8: Geographic Substructure in Pre-Nuragic Ancient Sardinia). In this new section, we divide our pre-Nuragic dataset into coarse geographic regions / sites (Supp. Mat 2) and use point symbols to reveal the geographic origins of individuals when projected onto the within Sardinia PCA plot. We show that the geographic regions do not vary along PC1 and PC2, as well as higher order PCs (1 way anova: $p_{PC1}=.5295$, $p_{PC2}=.752$, $p_{PC3}=.8072$, $p_{PC4}=.9805$) except for PC6 of the within Sardinia data ($p_{PC6}=.0435$). We also computed linear models of PC values versus latitude, longitude, and age and only found a significant association between PC6, longitude and age [See Supplemental Info Section 8]. Finally, we show the higher order PCs for reference. Overall, we cannot detect any strong and significant signal substructure amongst pre-Nuragic ancient individuals in the PCA projections on to modern Sardinian PCs. For thoroughness, we also repeat the same analysis for the projection onto the PCs used in our main Figure 2, and again do not find signatures of structure in the projections.

R3.1b : Given the low coverage of many of the Sardinian individuals, we do not expect such a pairwise distance metric to be computable with sufficient accuracy to enable clustering. For instance, the median proportion of missing sites across pairs of individuals is 0.862 with the 5th and 95th percentiles being 0.422 and 0.989. Such a large number of pairs with a low number of variants suggests clustering approaches on the ancient DNA alone will not work well. We now include this discussion in our new supplemental section.

Post-Nuragic admixture

[R3.2] The comparison with modern Sardinians suggests moderate levels of post-Nuragic admixture, with >40% non-Nuragic contribution even in the more isolated Ogliastro populations. This turnover is also reflected on the Y-chromosome, with the major ancient haplogroups I2-M223 and R1b-V88 much rarer among contemporary Sardinians. Overall, I enjoyed the

discussion of these results in the context of affinities with early farmers, and their tying together with similar results observed in Basques, which provides a nice resolution of some long-standing questions in European prehistory. Nevertheless, the amount of turnover across the entire island is somewhat surprising, so I would be interested to see whether any more insights could be made into its timing. Given that Nuragic individuals are good proxy population for at least the Sardinian source population, a possibility would be to test whether e.g. ALDER would yield an admixture LD curve for dating it using modern Sardinians as targets.

We agree the timing of post-Nuragic gene flow onto the island is an interesting question, and we now address this issue more directly by adding new samples in a new section “From the Nuragic to present-day Sardinia: Refined signatures of admixture and heterogeneity” and discussing the implications of these results for timing in the discussion.

Minor edits

[R3.3] Figure 1 – Including the names of sampling locations for the ancient individuals in the inset would be helpful.

We now depict the locality of each ancient Sardinian individual in a detailed plot at the beginning of Supp. Info. 1 (archaeological site descriptions). This should give the interested reader an overview over the geographic distribution of our sampling effort. We have added a reference to this supplemental figure to the Figure 1 caption.

[R3.4] Supplementary information 4 – The text mentions the “Sardinian” haplogroup I2-M26, but in figure S3 this is labelled as I2-L160.

Thank you for pointing out that inconsistency in reporting that clade. We now updated Figure S3 (now Figure S6) to show the values for I2-M26. The clade I2-L160 is a subclade of I2-M26. Almost all present-day Sardinians belong to it, so there is little difference in results other than the name change.

[R3.5] Fig S15 - The colours in the legend are mislabelled/switched for WHG (purple) and Middle East (red)

Thank you for spotting that reversion of labels. We have updated the figure caption to resolve this issue.

Reviewers' Comments:

Reviewer #1:

Remarks to the Author:

The authors have taken into consideration all the suggestions raised by the reviewers. I think the paper is now ready for publication in Nature Communications.

Reviewer #2:

Remarks to the Author:

I thank the authors for carefully revising their manuscript (and for nicely tracking changes in Latex). I recommend this article for publication. I think it is a really nice study, well written with a great level of nuance and detail making this an important example of an archaeogenomic project.

Reviewer #3:

Remarks to the Author:

In their revised manuscript, the authors have addressed my previous comments by performing additional analyses and including a new set of ancient samples from the post-Nuragic period. Their responses are exceptionally thorough, and the newly presented data provides intriguing new insights into the post-Nuragic population history of Sardinia. As such I don't have any further major comments, and would like to congratulate the authors on a substantial further strengthening of their study.

I only spotted two minor typos/mislabellings

Section "Post-Nuragic admixture of present-day Sardinians"

"We chose Cagliari, as PCA analysis revealed that it is a relatively homogeneous population in the center of present-day Sardinian variation (Fig. 2, main text)"

The newly revised Figure 2 does not appear to contain province labels in the PCA anymore.

Section "Refinement of admixture models using post-Nuragic ancient samples"

Numbering missing from references to "Section XX" and file "xx.csv"

REVIEWERS' COMMENTS:

Reviewer #1 (Remarks to the Author):

The authors have taken into consideration all the suggestions raised by the reviewers. I think the paper is now ready for publication in Nature Communications.

Reviewer #2 (Remarks to the Author):

I thank the authors for carefully revising their manuscript (and for nicely tracking changes in Latex). I recommend this article for publication. I think it is a really nice study, well written with a great level of nuance and detail making this an important example of an archaeogenomic project.

Reviewer #3 (Remarks to the Author):

In their revised manuscript, the authors have addressed my previous comments by performing additional analyses and including a new set of ancient samples from the post-Nuragic period. Their responses are exceptionally thorough, and the newly presented data provides intriguing new insights into the post-Nuragic population history of Sardinia. As such I don't have any further major comments, and would like to congratulate the authors on a substantial further strengthening of their study.

I only spotted two minor typos/mislabellings

Section "Post-Nuragic admixture of present-day Sardinians"

"We chose Cagliari, as PCA analysis revealed that it is a relatively homogeneous population in the center of present-day Sardinian variation (Fig. 2, main text)"

RESPONSE: Done.

The newly revised Figure 2 does not appear to contain province labels in the PCA anymore.

RESPONSE: The province labels occur now in Figure 5A, which is a zoom-in of the relevant region in Figure 2. We apologize for the confusion - but we think that this revised version is an improvement in allowing a more careful inspection of the relevant space of the PCA. We now added a reference to this zoom-in in the caption of Figure 2.

Section "Refinement of admixture models using post-Nuragic ancient samples"

Numbering missing from references to "Section XX" and file "xx.csv"

RESPONSE: We fixed the two missing references in the Supplementary Notes.